# Predictive Performance of Deep Quantum Data Re-uploading Models

**Xin Wang** [1]  **Han-Xiao Tao** [1]  **Re-Bing Wu** [1]

## Abstract

Quantum machine learning models incorporating data re-uploading circuits have garnered significant attention due to their exceptional expressivity and trainability. However, their ability to generate accurate predictions on unseen data, referred to as the predictive performance, remains insufficiently investigated. This study reveals a fundamental limitation in predictive performance when deep encoding layers are employed within the data re-uploading model. Concretely, we theoretically demonstrate that when processing high-dimensional data with limited-qubit data re-uploading models, their predictive performance progressively degenerates to near random-guessing levels as the number of encoding layers increases. In this context, the repeated data uploading cannot mitigate the performance degradation. These findings are validated through experiments on both synthetic linearly separable datasets and real-world datasets. Our results demonstrate that when processing high-dimensional data, the quantum data re-uploading models should be designed with wider circuit architectures rather than deeper and narrower ones.

## 1. Introduction

Quantum Machine Learning (QML) (Biamonte et al., 2017) has emerged as a promising field that integrates machine learning with quantum computing, offering potential computational advantages (Ristè et al., 2017; Huang et al., 2021b; Zhong et al., 2024). In recent years, quantum machine learning models based on Parameterized Quantum Circuits (PQCs) (Benedetti et al., 2019) have garnered significant attention (Yan et al., 2023; Meyer et al., 2023; Zhao et al., 2024). Although QML demonstrates advantages in

---

[1]Department of Automation, Tsinghua University, Beijing, China. Correspondence to: ReBing Wu <rbwu@tsinghua.edu.cn>.

*Proceedings of the 42nd International Conference on Machine Learning*, Vancouver, Canada. PMLR 267, 2025. Copyright 2025 by the author(s).

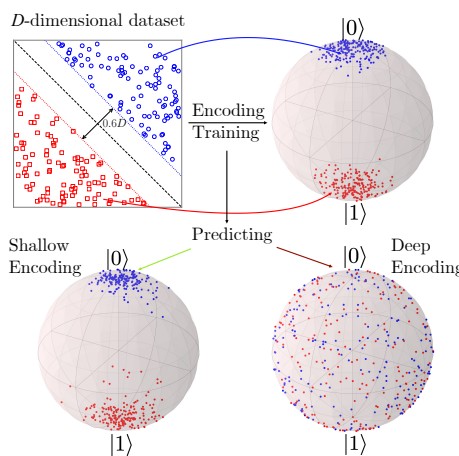

*Figure 1.* For $D$-dimensional linearly separable data, data re-uploading encodes the data into quantum circuits and trains the model effectively. However, during prediction, data re-uploading with shallow encoding layers maintains prediction results close to the training outcomes, while deep encoding layers lead to predictions that approach random guessing.

processing quantum data (Huang et al., 2021b; 2022), current experiments on classical data are primarily restricted to simple, low-dimensional datasets (Schuld et al., 2020; Hubregtsen et al., 2022; Pérez-Salinas et al., 2020; Huang et al., 2021a; Xu et al., 2024). The potential for achieving a quantum advantage in the processing of practical high-dimensional classical data remains to be demonstrated (Schuld & Killoran, 2022).

Encoding classical data into quantum states presents a significant challenge in QML (Lloyd et al., 2020; Wiebe, 2020; Weigold et al., 2021; Rath & Date, 2024). Currently, two primary encoding paradigms exist in supervised quantum machine learning: the encoding-variational paradigm and the data re-uploading paradigm (Jerbi et al., 2023). The encoding-variational paradigm encodes data into quantum states using methods such as basis encoding, angle encoding, or amplitude encoding, followed by training with parameterized quantum circuits (Schuld & Petruccione, 2018). In contrast, data re-uploading interleaves trainable parameterized gates between encoding gates, enabling more flexible data processing and enhancing model expressivity (Pérez-Salinas et al., 2020).

The encoding-variational paradigm suffers from several trainability issues, including encoding-induced optimization challenges (Li et al., 2022), exponentially numerous local minima in shallow circuits (You & Wu, 2021; Anschuetz & Kiani, 2022), and barren plateaus in deep circuits (Mc-Clean et al., 2018; Ragone et al., 2024). In contrast, data re-uploading improves trainability through interleaved parameterized and encoding gates, enhancing model expressivity (Pérez-Salinas et al., 2020). This approach supports deep circuit training and enables single-qubit circuits to approximate multivariate functions given sufficient depth (Pérez-Salinas et al., 2024).

The primary goal of machine learning is to achieve accurate predictions on previously unseen data. Researchers usually study trainability (training data performance) and generalization (training vs prediction gap) separately (Mohri, 2018). But good trainability or generalization alone doesn't guarantee good predictions. While encoding-variational QML shows good generalization (Caro et al., 2022), yielding small generalization error with few training samples. However, the predictive performance of data re-uploading models remain less understood. In contrast to prior work, we directly analyze how these models perform on unseen data and establish that, when the number of qubits is limited and the data dimension is high, their predictive performance degenerate to nearly random guessing.

## 1.1. Contributions

As shown in Fig. 1, we demonstrate that, regardless of the quality of training, the predictive performance of data re-uploading models on new data asymptotically approaches random guessing when the encoding circuit is deep. This typically occurs when the dimension of the data vectors far exceeds the number of qubits in the quantum circuits.

Our theoretical analysis shows that as encoding layers increase, predictions from data re-uploading circuits approach those from maximally mixed states, and repeated uploading cannot mitigate this problem.

We introduce a novel method to analyze the prediction error, bypassing the traditional decomposition into training and generalization errors. We directly analyze the expected output of the model over the data distribution. Our results demonstrate that when using data re-uploading models with deep encoding layers, the model's performance on the unseen new data approaches random guessing, regardless of how good the training results are, and regardless of the choice of loss function, optimization method (gradient-based or gradient-free), number of iterations, model parameter count, or training sample size. After decomposing prediction error into training error and generalization error, increasing model complexity can reduce training error but might increase generalization error, while increasing sam-

ple size can reduce generalization error but might increase training error. With these dynamic changes, it's difficult to determine the resulting prediction error (their sum), whereas our approach bypasses this limitation by directly analyzing the prediction error.

To establish this phenomenon, we analyze the model's predictive performance by examining the expected output over the data distribution. Our analysis focuses on two key aspects: the impact of the number of encoding layers and the number of repetitions. For the former, we employ techniques similar to Li's paper (Li et al., 2022), which originally only permitted specific non-parameterized entangling gates (CNOT or CZ) between encoding layers. We extend these techniques by allowing arbitrary learnable parameterized quantum gates between encoding layers. For the latter, Li's techniques cannot analyze repeated data uploading scenarios, we address this limitation by constructing approximating circuits to analyze cases with repeated data uploads. Importantly, repeated data uploading is crucial, as it significantly enhances the trainability of data re-uploading models (Pérez-Salinas et al., 2020; 2024; Yu et al., 2022; 2023). Furthermore, the data re-uploading paradigm with its good trainability addresses the trainability issues identified in Li's paper, and our work further extends this research by focusing on analyzing the predictive capabilities of these models.

Our experiments confirm the theory, showing that differences in encoding layers notably affect predictive performance, despite identical datasets, parameter counts, and training errors. Additionally, we explain why data re-uploading models perform well on MNIST (LeCun) even with deep encoding layers.

These findings offer critical insights into the architectural design of data re-uploading models, indicating that the accurate prediction of high-dimensional data is unlikely to be accomplished by few-qubit quantum circuits, notwithstanding their trainability and expressivity.

## 1.2. Related Work

**Quantum Encoding** The properties of quantum encoding strategies has investigated several fundamental aspects. (Schuld et al., 2021) examined how different encoding strategies affect the expressivity of models. Regarding robustness, (LaRose & Coyle, 2020) indicated that encoding impacts the robustness of quantum classifiers against quantum noise. Regarding generalization, previous research (Caro et al., 2021) derived generalization bounds related to different encoding strategies for quantum models. Another study (Li et al., 2022) demonstrated that for angle encoding, the average quantum state exponentially approaches the maximally mixed state as the encoding layers increases.

**Data Re-uploading** The exploration of data re-uploading has predominantly focused on its trainability. The concept was initially introduced in (Pérez-Salinas et al., 2020), where the advantageous trainability properties were experimentally demonstrated. Subsequent theoretical work by (Pérez-Salinas et al., 2024) established that increasing depth enables single data re-uploading models to approximate any multivariate function. Further theoretical foundations were provided by (Yu et al., 2022; 2023), who rigorously proved these models' advantages in function approximation compared to classical ReLU neural networks. (Barthe & Pérez-Salinas, 2024) augmented these findings with a detailed analysis of gradient behavior and frequency profiles. These benefits have led to widespread use in quantum machine learning, typically with shallow encoding layers and multiple repetitions (Dutta et al., 2022; Wach et al., 2023; Cassé et al., 2024; Rodriguez-Grasa et al., 2024), or on sparse datasets (Periyasamy et al., 2022; Jerbi et al., 2023). Regarding generalization capability, (Zhu et al., 2025) established the relationship between the number of data re-uploading repetitions, training epochs, and the model's generalization performance.

**Predictive Performance of QML:** The investigation into the predictive performance of quantum machine learning models has been relatively scarce. (Caro et al., 2022) derived generalization error bounds for encoding-variational QML. (Wang et al., 2024) combined these generalization error bounds with the training error bounds of AdaBoost to establish prediction error bounds. In quantum kernel methods, (Wang et al., 2021) established generalization error under conditions of quantum noise. For data re-uploading models, the predictive performance has not been well understood.

## 2. Preliminaries

### 2.1. Quantum Computing

We first introduce elementary concepts in quantum computing. The state space of an $N$-qubit quantum system is a $2^N$-dimensional complex Hilbert space $\mathcal{H} \cong \mathbb{C}^{2^N}$. The computational basis for this Hilbert space consists of $\{|0\rangle, |1\rangle, ..., |2^N - 1\rangle\}$, where each basis state $|i\rangle$ represents a unique binary string of length $2^N$ corresponding to the binary representation of integer $i$.

Information in quantum systems is stored in quantum states, which can be represented by a positive semi-definite matrix $\rho \in \mathbb{C}^{2^N \times 2^N}$ with property $\text{Tr}[\rho] = 1$. If $\text{Tr}[\rho^2] = 1$, the quantum state is called a pure state; otherwise, it is a mixed state. For $N$-qubit pure states, they can be represented by a unit state vector $|\varphi\rangle \in \mathbb{C}^{2^N}$, where $\rho = |\varphi\rangle \langle\varphi|$ and $\langle\varphi| = |\varphi\rangle^\dagger$. Mixed states are convex combinations of pure states. In particular, for an $N$-qubit system, the maximally

mixed state $\rho_I = \frac{I}{2^N}$ represents a state with no information.

Quantum states evolve through quantum circuits (mathematically represented as unitary transformations $U$), transforming from $\rho$ to $\rho'$ according to $\rho' = U\rho U^\dagger$. Quantum circuits are composed of quantum gates, with typical single-qubit gates being rotation gates of the form $R_P(\phi) = e^{-i\phi P/2}$, where $P \in \{X, Y, Z\}$ are the Pauli matrices as follows:

$$X = \begin{pmatrix} 0 & 1 \\ 1 & 0 \end{pmatrix}, Y = \begin{pmatrix} 0 & -i \\ i & 0 \end{pmatrix}, Z = \begin{pmatrix} 1 & 0 \\ 0 & -1 \end{pmatrix}.$$

In particular, any single-qubit gate can be represented as $R(\phi_1, \phi_2, \phi_3) = R_z(\phi_3) R_y(\phi_2) R_z(\phi_1)$. Multi-qubit gates, also known as entangling gates, entangle multiple qubits together, with the CNOT gate and CZ gate being typical examples. To extract information from quantum circuits for classical processing, measurements are performed using Hermitian matrices $H$, with the expectation value given by $\text{Tr}[H\rho]$, where $H$ is called the observable.

### 2.2. Data Re-uploading

Data re-uploading was first proposed by (Pérez-Salinas et al., 2020), with its core idea being to enhance PQC-based quantum classifiers by alternately constructing data loading gates and data processing gates. In (Pérez-Salinas et al., 2020), both the single-qubit data encoding gates and single-qubit data processing gates take the form:

$$\begin{aligned} R(\boldsymbol{\phi}) &= R(\phi_1, \phi_2, \phi_3) = R_z(\phi_3) R_y(\phi_2) R_z(\phi_1) \\ &= e^{-i\phi_3 Z/2} e^{-i\phi_2 Y/2} e^{-i\phi_1 Z/2}, \end{aligned} \tag{1}$$

where $Z$ and $Y$ are Pauli matrices. The gate $R(\boldsymbol{\phi})$ encodes data when $\boldsymbol{\phi} = \boldsymbol{x}$ and processes data when $\boldsymbol{\phi} = \boldsymbol{\theta}$. In multi-qubit circuits, entangling gates (e.g., CNOT, CZ) link individual qubits to generate quantum entanglement.

In this paper, we consider a more general architecture of data re-uploading circuit. The circuit loads data exclusively through single-qubit encoding gates as defined in Eq. (1), while allowing arbitrary parameterized unitary gates. Given that the limited-scale quantum circuit has $N$ qubits, the entire data whose dimension is larger than $3N$ has to be uploaded via multiple encoding layers. A simple way is to divide the data into $L$ chunks and upload chunks via $L$ encoding layers, where $L \geqslant \lceil \frac{D}{3N} \rceil$. This data is then repeatedly uploaded into the circuit $P$ times. The complete architecture of the quantum data re-uploading circuit is illustrated in Fig. 2.

Without loss of generality, we assume that the dimension of data $\boldsymbol{x}$ is $D = 3NL$, the entire data is divided into $L$ chunks $\boldsymbol{x} = [\boldsymbol{x}_{[1]}, \boldsymbol{x}_{[2]}, \cdots, \boldsymbol{x}_{[L]}]$, each chunk $\boldsymbol{x}_{[l]} \in \mathbb{R}^{3N}$ contains $N$ three-dimensional data $\boldsymbol{x}_{l,n} = [x_{l,n,1}, x_{l,n,2}, x_{l,n,3}]$ and every $\boldsymbol{x}_{l,n}$ is encoded by the single-qubit encoding gate

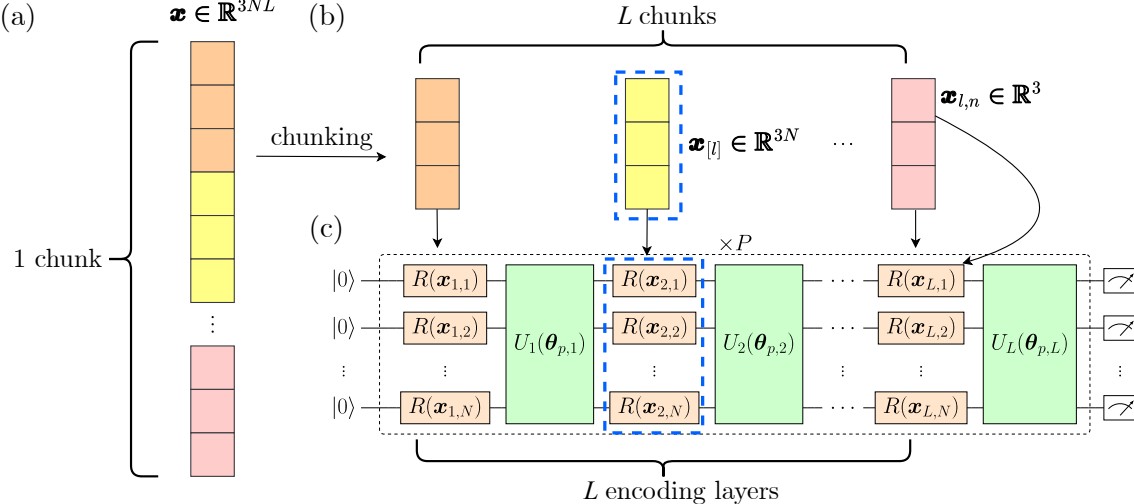

*Figure 2.* Data re-uploading encoding process. (a) The original data. (b) Divide original data into $L$ chunks. (c) Each data chunk is encoded by an encoding layer. The entire data is re-uploaded into the circuit $P$ times, where the parameterized gates in each repetition can be arbitrary and can differ between repetitions.

$R(\boldsymbol{x}_{l,n})$ in the $l$-th layer and $n$-th qubit. $l$-th layer's encoding gate is denoted as $R_l(\boldsymbol{x}_{[l]})$. Following (Li et al., 2022), we assume data elements follow independent Gaussian distributions $x_{l,n,i} \sim \mathcal{N}(\mu_{l,n,i}, \sigma^2_{l,n,i})$.

The parameterized gates after $l$-th encoding layer in $p$-th repetition is denoted as $U_l(\boldsymbol{\theta}_{p,l})$. Note that each repetition uses the same data vector but with different parameter vectors. Let $\boldsymbol{\theta} = [\boldsymbol{\theta}_{[1]}, \cdots, \boldsymbol{\theta}_{[P]}]$ represent the collection of all parameters, then the quantum gate of $P$-repeated data re-uploading circuit can be expressed as:

$$V(\boldsymbol{x}, \boldsymbol{\theta}) = \prod_{p=1}^{P} \overleftarrow{\prod_{l=1}^{L}} U_l(\boldsymbol{\theta}_{p,l}) R_l(\boldsymbol{x}_{[l]}),$$

where $\overleftarrow{\prod}_{l=1}^{L} A_l = A_L A_{L-1} \cdots A_1$.

Data re-uploading circuits starts from initial state $\rho_0 = |0\rangle\langle 0|$, the quantum state evolves through $P$-repeated data re-uploading circuit $V(\boldsymbol{x}, \boldsymbol{\theta})$. The expectation value of measurement on the evolved state with respect to an observable $H$ is:

$$h_P(\boldsymbol{x}) = \text{Tr}\left[HV(\boldsymbol{x}, \boldsymbol{\theta})\rho_0 V(\boldsymbol{x}, \boldsymbol{\theta})^\dagger\right].$$

Different tasks require different observable $H$. In this paper, the eigenvalues of observable lies in $[-1, 1]$.

### 2.3. Supervised Quantum Machine Learning

In supervised machine learning, we consider a dataset $S = \{(\boldsymbol{x}^{(m)}, y^{(m)})\}_{m=1}^{M}$ containing $M$ samples. Each sample $(\boldsymbol{x}^{(m)}, y^{(m)})$ consists of a feature vector $\boldsymbol{x}^{(m)}$ and

its corresponding label $y^{(m)}$, independently drawn from a same distribution. Here, $\mathcal{X}$ denotes the feature space and $\mathcal{Y}$ represents the label space. This paper focuses on the deterministic scenario where each feature $\boldsymbol{x}$ sampled from $\mathcal{D}_\mathcal{X}$ has a unique label $y(\boldsymbol{x})$. The machine learning model learns a hypothesis $h_S$ from dataset $S$, aiming to correctly predict labels for new features drawn from $\mathcal{D}_\mathcal{X}$.

In this paper, we consider both classification and regression tasks. For classification tasks, we focus on binary classification where the labels corresponding to $\boldsymbol{x}$ are $y(\boldsymbol{x}) \in \{0, 1\}$. We employ $H_{y(\boldsymbol{x})}$ as the observable for feature $\boldsymbol{x}$, where $H_0 = |0\rangle\langle 0|$ and $H_1 = |1\rangle\langle 1|$ are single-qubit observables acting on the first qubit. Since the measurement results for both $H_0$ and $H_1$ are greater than 0 and sum to 1, these results can be interpreted as probabilities of belonging to each class. The hypothesis function learned by $P$-repeated data re-uploading models trained on dataset $S$ is given by $h_S(\boldsymbol{x}) = h_P(\boldsymbol{x}, \boldsymbol{\theta}^*)$ with observable $H_{y(\boldsymbol{x})}$, where $\boldsymbol{\theta}^*$ is the parameters chosen during training. $h_S(\boldsymbol{x})$ indicates the probability that feature $\boldsymbol{x}$ belongs to the correct class.

For regression tasks, we consider target functions $f(\boldsymbol{x}) \in [-1, 1]$ with observable $H_L = \bigotimes_{n=1}^{N} Z_n$. The hypothesis $h_S(\boldsymbol{x})$ represents the predicted value of $f(\boldsymbol{x})$.

To evaluate the predictive performance of hypothesis $h_S$, we define its prediction error on distribution $\mathcal{D}_\mathcal{X}$ for classification tasks as:

$$R^C(h_S) = \mathop{\mathbb{E}}_{\boldsymbol{x} \sim \mathcal{D}_\mathcal{X}} [|1 - h_S(\boldsymbol{x})|],$$

This prediction error evaluates the gap between the probability that hypothesis $h_S$ predicts the feature as the correct

class and 1.

For new features, class prediction is performed by measuring observables $H_0$ and $H_1$, then assigning the class with the higher measurement result. Crucially, In this paper, we assume that each feature $\boldsymbol{x}$ inherently possesses a unique true label $y(\boldsymbol{x})$. This allows performance evaluation by comparing the model's probability of correct prediction against 1 (representing perfect certainty). If the prediction probability approaches random guessing levels(e.g., near 0.5 in binary classification), this statistically indicates model failure to learn meaningful patterns, even without label knowledge of new features.

For regression tasks, we define the prediction error to evaluate the gap between the output of $h_S$ and $f(\boldsymbol{x})$ as:

$$R^L(h_S) = \mathbb{E}_{\boldsymbol{x} \sim \mathcal{D}_{\mathcal{X}}}[|f(\boldsymbol{x}) - h_S(\boldsymbol{x})|].$$

Meanwhile, we also care about the hypothesis's performance on the training set $S = \{(\boldsymbol{x}^{(m)}, y^{(m)})\}_{m=1}^M$, defining the training error for classification as:

$$\widehat{R}_S^C(h_S) = \frac{1}{M} \sum_{m=1}^M \left|1 - h_S(\boldsymbol{x}^{(m)})\right|,$$

Similarly, for regression tasks, we define the training error as:

$$\widehat{R}_S^L(h_S) = \frac{1}{M} \sum_{m=1}^M \left|f(\boldsymbol{x}^{(m)}) - h_S(\boldsymbol{x}^{(m)})\right|.$$

Since the prediction error cannot be directly computed, it is typically analyzed by decomposing it into two components: the directly computable training error and the generalization error $\text{gen}(h_S)$. For classification tasks, this decomposition is expressed as $\text{gen}(h_S) = R^C(h_S) - \widehat{R}_S^C(h_S)$ (regression problems follow the same principle). In practical applications, we estimate the prediction error by evaluating the model's performance on a test set.

### 2.4. Quantum Divergence

Similar to the Kullback-Leibler divergence in classical information theory, the quantum Petz-Rényi divergence ([Petz](#), 1986) measures the distinguishability between two quantum states:

$$D_\alpha(\rho_1 \| \rho_2) \coloneqq \frac{1}{\alpha - 1} \log_2 \left( \text{Tr} \left[ \rho_1^\alpha \rho_2^{1-\alpha} \right] \right), \quad (2)$$

where $\alpha \in (0, 1) \cup (1, +\infty)$. While not a formal distance metric, it enables critical quantum hypothesis testing analyses ([Audenaert](#), 2007; [Nussbaum & Szkoła](#), 2009). For analyzing quantum encoded states, we employ $\alpha = 2$:

$$D_2(\rho_1 \| \rho_2) = \log_2 \left( \text{Tr} \left[ \rho_1^2 \rho_2^{-1} \right] \right), \quad (3)$$

which has important applications in both quantum machine learning ([Liu et al.](#), 2022) and quantum communication ([Fang & Fawzi](#), 2021). The $D_2$ divergence can effectively analyze quantum state distinguishability in the Pauli basis, bounding observable measurement differences. (see App. [B](#)). For any $N$-qubit quantum states $\rho$ and maximally mixed state $\rho_I$, $D_2(\rho \| \rho_I) \in [0, N]$ (see App. [F](#)), reaching its minimum value of 0 if and only if $\rho = \rho_I$. In this paper, when referring to the quantum divergence, we mean the $D_2$ divergence defined in Eq. [(3)](#).

## 3. Main Results

This section presents our main theoretical findings on how encoding layers and repetitions affect the predictive performance of data re-uploading models. Subsec. [3.1](#) demonstrates exponential approaching of expected encoded states to maximally mixed states with encoding layers. Subsec. [3.2](#) shows the ineffectiveness of repeated uploading. Subsec. [3.3](#) derives theoretical bounds for prediction error.

### 3.1. Dependence on the Number of Encoding Layers

We first analyze how encoding layers affect the expected encoded state generated by single-upload ($P = 1$) data re-uploading circuits. The encoding transformation of input data $\boldsymbol{x} \mapsto R(\boldsymbol{x})\rho R(\boldsymbol{x})^\dagger$ introduces nonlinearity that complicates direct analysis of expected quantum states. To overcome this, we adopt a Pauli basis decomposition approach, examining the evolution of Pauli coefficients of expected quantum states within this framework.

For an $N$-qubit quantum state $\rho$, its representation in the Pauli basis can be expressed as:

$$\rho = \frac{1}{2^N} \left( \sum_{P_i \in \{I, X, Y, Z\}^{\otimes N}} \alpha_i P_i \right), \quad (4)$$

where $\alpha_i = \text{Tr}\left[\rho P_i\right]$ is the coefficient for Pauli basis element $P_i$. Since each $P_i$ is Hermitian, all coefficients $\alpha_i$ are real numbers, with $4^N$ terms in total. Under unitary transformation $U$, the state $U\rho U^\dagger$ keeps the same Pauli basis structure, only changing the coefficients $\alpha_i$. Our analysis will focus on these coefficient changes.

When the encoding gate $R(\boldsymbol{x})$ is applied to a single-qubit state $\rho$, where $\boldsymbol{x} = [x_1, x_2, x_3]$ follows independent Gaussian distributions $x_i \sim \mathcal{N}(\mu_i, \sigma_i^2)$ with $\sigma_i^2 > \sigma^2, i \in [3]$, we can analyze the state by decomposing it in the Pauli basis $\{I, Z, X, Y\}$ with corresponding coefficients $[\alpha_I, \alpha_Z, \alpha_X, \alpha_Y]$. Considering the expected state after encoding $\mathbb{E}[\rho] = \mathbb{E}_{\boldsymbol{x}}[R(\boldsymbol{x})\rho_0 R(\boldsymbol{x})^\dagger]$, let $[\beta_I, \beta_Z, \beta_X, \beta_Y]$ denote the Pauli basis coefficients of $\mathbb{E}[\rho]$. We establish that $\beta_Z^2 + \beta_X^2 + \beta_Y^2 \leqslant e^{-\sigma^2}(\alpha_Z^2 + \alpha_X^2 + \alpha_Y^2)$ while $\beta_I = \alpha_I$ remains unchanged. This demonstrates that the magnitude

of Pauli coefficients for non-identity terms decays exponentially with the variance $\sigma^2$. As the encoding layers $L$ increases, these non-identity coefficients rapidly approach zero, leaving only the identity term. Consequently, the quantum state converges to the maximally mixed state. The following theorem formalizes this phenomenon.

**Theorem 3.1.** *Consider an $N$-qubit data re-uploading circuit with $L$ encoding layers and without repetition ($P = 1$), which encodes data $\boldsymbol{x} \in \mathbb{R}^{3NL}$ into the circuit, where each data point follows an independent Gaussian distribution, i.e., $x_{l,n,i} \sim \mathcal{N}(\mu_{l,n,i}, \sigma^2_{l,n,i})$ and $\sigma^2_{l,n,i} \geqslant \sigma^2$. Let $\rho(\boldsymbol{x}, \boldsymbol{\theta})$ denote the $N$-qubit encoded state. Then the quantum divergence between the expected state $\mathbb{E}[\rho] = \mathbb{E}_{\boldsymbol{x}}[\rho(\boldsymbol{x}, \boldsymbol{\theta})]$ and the maximally mixed state $\rho_I = \frac{I}{2^N}$ satisfies:*

$$D_2\left(\mathbb{E}[\rho]||\rho_I\right) \leqslant \log_2\left(1 + (2^N - 1)e^{-L\sigma^2}\right).$$

Intuitively, when quantum states become indistinguishable, their measurement results for observables with eigenvalues in $[-1, 1]$ also become identical. Theorem 3.1 shows this occurs when $L \geqslant \frac{1}{\sigma^2}[(N + 2)\ln 2 + 2\ln(\frac{1}{\epsilon})]$, $|\mathbb{E}_{\boldsymbol{x}}[h_S(\boldsymbol{x})] - h_I| \leqslant \epsilon$, where $\mathbb{E}_{\boldsymbol{x}}[h_S(\boldsymbol{x})] = \text{Tr}\,[H\mathbb{E}[\rho]]$ and $h_I = \text{Tr}\,[H\rho_I]$. This means that the expectation of hypothesis $h_S$ with respect to feature $\boldsymbol{x}$ is nearly informationless (see App. C).

It is worth noting that Theorem 3.1 extends beyond the scope of Theorem 2 in (Li et al., 2022) by allowing arbitrary learnable parameterized quantum gates between encoding layers, rather than being limited to specific non-parameterized entangling gates.

### 3.2. Dependence on the Number of Repetitions

Due to repeated data uploading, even the Pauli coefficients of quantum states have nonlinear relationships with input data, making direct analysis methods inapplicable. To address this, we approximate the data re-uploading circuits with $P > 1$ using an approximating circuit.

For a data vector $\boldsymbol{x} = [x_1, \cdots, x_D] \in \mathbb{R}^D$, we focus on a single element $x_d \in [0, 2\pi]$ (periodicity follows from Eq. (1)). We expand $x_d$ in binary form as $\sum_{j=-3}^{q} b_j 2^{-j} + \epsilon_q$, where $b_j \in \{0, 1\}$ and $|\epsilon_q| \leqslant 2^{-q}$. Here, the first 3 bits encode the integer part and $q$ bits encode the fractional part. Let $\tilde{x}_d$ denote the approximation of $x_d$ with $q + 3$ bits.

To approximate $R_z(x_d)$, we use working qubits (for final measurement) as targets and auxiliary qubits (for encoding data $\tilde{x}_d$) as controls. Controlled rotations $C - R_z(2^{-j})$ are applied with auxiliary qubits $|b_j\rangle$ as controls and working qubits as targets, approximating $R_z(x_d)$ with $R_z(\tilde{x}_d)$. This approach naturally extends to $R_y(x)$ gates.

For an $N$-qubit data re-uploading circuit with $L$ encoding

layers and $P$ repetitions, we construct an approximating circuit using $N + 3NL(q+3)$ qubits (Fig. 3). Let $h_P(\boldsymbol{x}, \boldsymbol{\theta})$ and $\tilde{h}_P(\boldsymbol{x}, \boldsymbol{\theta})$ denote the outputs of the original and approximating circuits respectively. The approximation error satisfies $|\tilde{h}_P(\boldsymbol{x}, \boldsymbol{\theta}) - h_P(\boldsymbol{x}, \boldsymbol{\theta})| \leqslant \delta$ when $q \geqslant \lceil \log_2(3PLN/\delta) \rceil$.

For both $P = 1$ and $P > 1$ circuits, we construct approximating circuits with outputs $\tilde{h}_1(\boldsymbol{x}, \boldsymbol{\theta})$ and $\tilde{h}_P(\boldsymbol{x}, \boldsymbol{\theta})$ respectively. The $P$-repeated approximating circuit essentially applies multiple data-independent unitary gates to the circuit with $P = 1$. Since unitary operations cannot enhance the distinguishability between quantum states, $P$ repetitions cannot improve the indistinguishability caused by deep encoding layers. That is, if $|\tilde{h}_1(\boldsymbol{x}, \boldsymbol{\theta}) - h_I| \leqslant \epsilon$, then $|\tilde{h}_P(\boldsymbol{x}, \boldsymbol{\theta}) - h_I| \leqslant \epsilon$. This leads to the following theorem for data re-uploading circuits with $P$ repetitions (see App. D).

**Theorem 3.2.** *Consider an $N$-qubit data re-uploading circuit with $L$ encoding layers and $P$ repetitions, which encodes data $\boldsymbol{x} \in \mathbb{R}^{3NL}$ into the circuit, where each data point follows an independent Gaussian distribution, i.e., $x_{l,n,i} \sim \mathcal{N}(\mu_{l,n,i}, \sigma^2_{l,n,i})$ and $\sigma^2_{l,n,i} \geqslant \sigma^2$. Let $h_P(\boldsymbol{x}, \boldsymbol{\theta})$ be its output with respect to an observable $H$ whose eigenvalues lie in $[-1, 1]$. When $L \geqslant \frac{1}{\sigma^2}[(N + 2)\ln 2 + 2\ln(\frac{1}{\epsilon})]$, we have:*

$$\left|\mathbb{E}_{\boldsymbol{x}}[h_P(\boldsymbol{x}, \boldsymbol{\theta})] - h_I\right| \leqslant \epsilon,$$

*where $h_I = \text{Tr}\,[H\rho_I]$ and $\rho_I = \frac{I}{2^N}$ is the $N$-qubit maximally mixed state.*

It is important to note that Theorem 3.2 extends to arbitrary repetition numbers, going beyond the non-repetition case addressed in Corollary 2.1 of (Li et al., 2022).

### 3.3. Bounds on the Prediction Error

We establish that data re-uploading models with deep encoding layers exhibit limitations in predictive performance:

**Proposition 3.3** (Informal). *When the conditions of Theorem 3.2 hold, the prediction error of the hypothesis $h_S$ generated by the data re-uploading model satisfies:*

*(1) For binary classification tasks: $\left|R^C(h_S) - \frac{1}{2}\right| < \epsilon$;*

*(2) For regression tasks: $R^L(h_S) \geqslant \left|\mathbb{E}_{\boldsymbol{x} \sim \mathcal{D}_{\mathcal{X}}}[f(\boldsymbol{x})]\right| - \epsilon$.*

For classification tasks, when $R^C(h_S) = \frac{1}{2}$, it indicates that the hypothesis $h_S$ assigns equal probability to new features belonging to either the correct or incorrect class. For regression tasks, the lower bound implies that the prediction error depends solely on the target function and is independent of the learned hypothesis $h_S$. In both cases, these results demonstrate that the model fails to extract useful information.

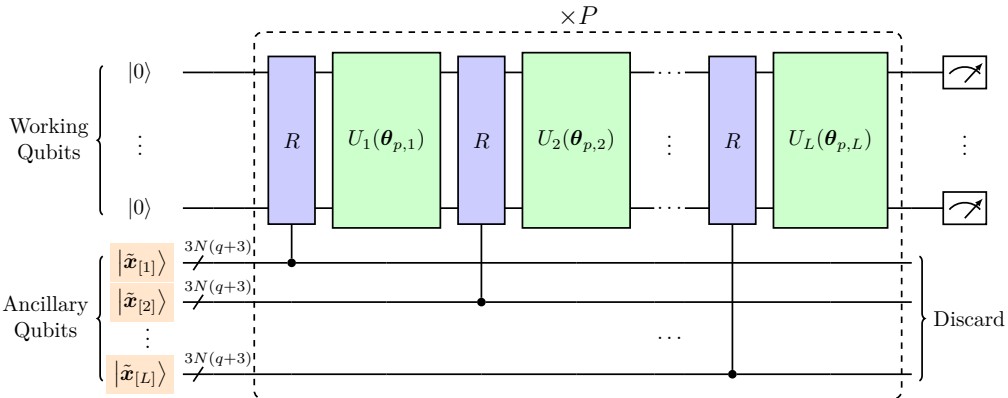

*Figure 3.* Approximating circuit for data re-uploading circuits. The qubits in approximating circuit are divided into two parts: working qubits and auxiliary qubits. Firstly, we use $3NL(q+3)$ ancillary qubits to encode the binary digits of feature $\boldsymbol{x} \in \mathbb{R}^{3NL}$ into the circuit. Then, we use the data-independent control gate $CR$ between the working qubit as target qubit and auxiliary qubits as control qubits to approximate the encoding gate in working qubits. Note that the approximating circuit is only used for theoretical analysis.

## 4. Experiments

### 4.1. Divergence Experiments

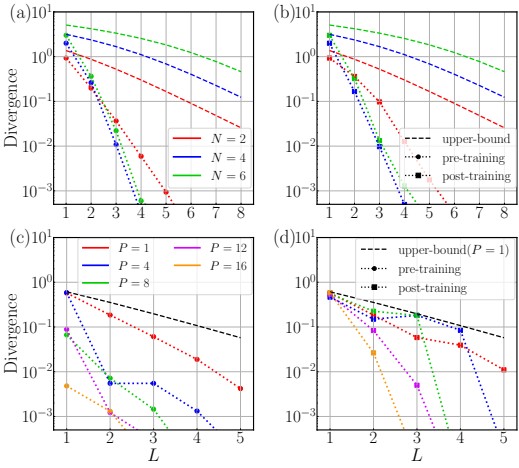

*Figure 4.* Divergence versus encoding layer $L$. Panels (a, b): varying qubit number $N$ at fixed repetition $P = 1$; panels (c, d): varying repetition $P$ at fixed qubit number $N = 1$. Pre-training (before training, in panels (a, c)) and post-training (after training, in panels (b, d)) results are compared with theoretical upper bounds. Panels (a, b) share common legends, and panels (c, d) share common legends: colors in panels (a, c) represent different $N$ or $P$, and line styles in panels (b, d) represent conditions (upper-bound, pre-training, post-training). Y-axis is logarithmic.

**Data:** To validate Theorem 3.1, we measure the divergence in binary classification datasets. Each dataset sample $(\boldsymbol{x}^{(m)}, y^{(m)})$ consists of a $D$-dimensional feature vector $\boldsymbol{x}^{(m)} \in \mathbb{R}^D$ and a class label $y^{(m)}$. The elements of $\boldsymbol{x}^{(m)}$ are independently drawn from Gaussian distributions, where the $d$-th element follows $\mathcal{N}(\mu_d, \sigma_d^2)$.

The mean values $\mu_d$ are class-dependent: for class one, $\mu_d = \left[\frac{2\pi}{16}(d \bmod 8)\right] \bmod 2\pi$, while for class two, $\mu_d = \left[\frac{2\pi}{16}(8 + (d \bmod 8))\right] \bmod 2\pi$. Each dimension $d$ has variance $\sigma_d^2 = 0.8$ for both classes.

**Experimental Setup:** We use a data re-uploading model as shown in Fig. G.1. The training set contains 2000 samples, and the test set contains 1,000,000 samples (see App. F). With normally distributed initial parameters, we trained models using cross-entropy loss with Adam optimizer (learning rate = 0.005) over 1000 epochs (batch size = 200), selecting parameters with the lowest training error for test. We computed divergence between maximally mixed state and average encoded states for both classes, then averaged these values as the final divergence result.

**Results:** For fixed repetition $P = 1$, we examine the dependence of quantum divergence on qubit number and encoding layers. As shown in Fig. 4(a), the divergence exhibits exponential decay with increasing encoding layers $L$. While training induces a marginal increase in divergence as shown in Fig. 4(b), this effect is negligible compared to the exponential decay trend.

For fixed qubit number $N = 1$, we examine the dependence of divergence on encoding layers and repetition. As shown in Fig. 4(c), without training, more repetitions may lead to faster decay of divergence. However, as shown in Fig. 4(d), after training, the divergence with multiple repetitions approaches the upper bound of $P = 1$ given by Theorem 3.1. Notably, when the number of encoding layer $L = 1$, the divergence remains close to the theoretical upper bound regardless of the number of repetitions, which explains why circuits with $L = 1$ perform well in quantum machine learning tasks.

## 4.2. Classification Experiments

**Data:** To demonstrate the limitations of quantum data re-uploading caused by deep encoding layers, we consider a simple linear separable classification problem. For $D$-dimensional data $\boldsymbol{x}$ distributed in $[-\frac{\pi}{2}, \frac{\pi}{2}]^D$, if $\mathbf{1}^\top \boldsymbol{x} > 0.3D$, the data belongs to class one, and if $\mathbf{1}^\top \boldsymbol{x} < -0.3D$, it belongs to class two, where $\mathbf{1}$ is a vector of ones.

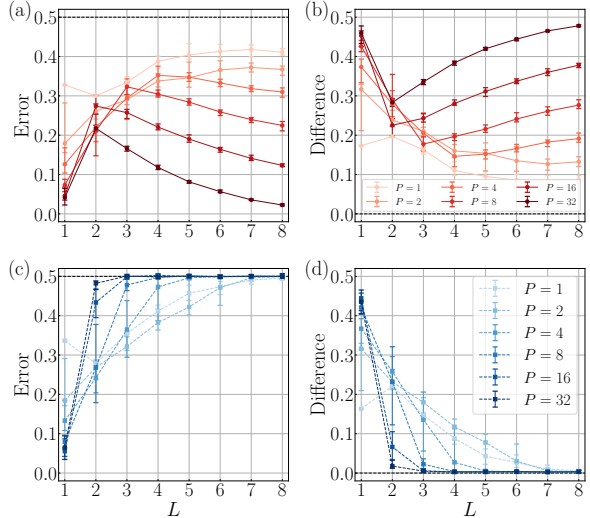

*Figure 5.* (a) Training error, (c) Test error, (b), (d) Difference between the model's output with respect to the observable $H_0$ on training data and test data compared to $\mathrm{Tr}\,[H\rho_I]$, respectively. Error bars represent the minimum and maximum values across 10 independent runs with different random seeds, with the central line showing the mean value.

**Experimental Setup:** We use a single-qubit data re-uploading model (Fig. G.1) for classification. A single qubit is chosen because it has received significant attention (Pérez-Salinas et al., 2020; Yu et al., 2022; Pérez-Salinas et al., 2024) and serves as a typical example for examining the limitations of deep data re-uploading models. Besides, the single-qubit model can exclude the influence of trainability. To eliminate the impact of parameter count, we conduct comparative experiments using quantum circuits with fixed total encoding layers $L_{\max} = 8$. For data with $L$ chunks, we maintain the total circuit layers at $L_{\max}$ by only encoding data in the first $L$ layers and setting the data input to zero vectors for the remaining layers (using $L$ encoding layers).

In numerical experiments, the training set contains 600 samples and the test set contains 10000 samples. With normally distributed initial parameters, we trained models using cross-entropy loss with Adam optimizer (learning rate = 0.005) over 1000 epochs (batch size = 200), selecting parameters with the lowest training error for test. The experiments were repeated 10 times with randomly initialized parameters.

**Results:** Fig. 5(a) and (b) show the training and test error of

the model with different encoding layers $L$ and repetitions $P$. As $L$ increases, the test error gradually increases to 0.5 (equivalent to random guessing). As $P$ increases, the training error decreases, but the test error converges to 0.5 faster with increasing $L$. Fig. 5(c) and (d) display the difference between the model's output with respect to the observable $H_0$ and $\mathrm{Tr}\,[H_0\rho_I]$ for training and test data, corresponding to Fig. 6(a) and (b). As the test error approaches 0.5, the difference on test data approaches zero, which verifies Theorem 3.2.

### 4.3. Classification Experiments in Same Dataset

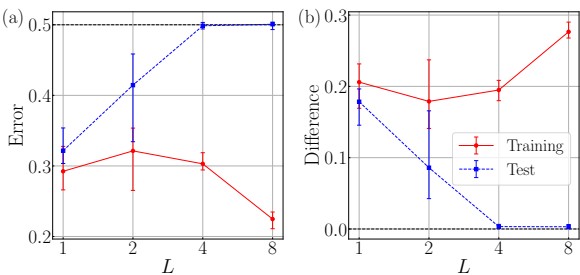

*Figure 6.* (a) Training and test error with same datasets, same repetition, but different encoding layers, (b) Difference between the output with respect to $H_0$ on training and test data and $\mathrm{Tr}\,[H\rho_I]$. Error bars represent the minimum and maximum values across 10 independent runs with different random seeds, with the central line showing the mean value.

**Data:** To eliminate the influence of data dimensionality on experimental results, we employed the same data as described in Subsec 4.2, with a feature dimension of $D = 24$.

**Experimental Setup:** We used data re-uploading circuits (Fig. G.1, excluding CNOT gates to prevent entanglement effects, as entanglement may degrade training and prediction performance (Ortiz Marrero et al., 2021; Leone et al., 2024)) with the same number of parameters: $(N = 1, L = 8)$, $(N = 2, L = 4)$, $(N = 4, L = 2)$, and $(N = 8, L = 1)$, all with $P = 8$ repetitions. Other experimental settings follow those in Subsec. 4.2.

**Results:** Fig. 6 shows as encoding layers $L$ increases, the test error gradually increases to 0.5, the corresponding difference approaches zero. Despite similar training errors in same dataset between $L = 1$ and $L = 4$, the test performance differed significantly due to encoding layers.

### 4.4. Real-world Dataset Experiments

**Dataset:** CIFAR-10-Gray (airplane/automobile, grayscale, 12×12 pixels), CIFAR-10-RGB (airplane/automobile, RGB, 12 × 12 pixels), MNIST (digit 0/1, 12 × 12 pixels).

**Experimental Setup:** We used circuits (Fig. G.1) with $N = 8$ and $P = 2$, using encoding layers $L = 6$ for CIFAR-

10-Gray and MNIST, and $L = 18$ for CIFAR-10-RGB. The training set contain 600 samples and the test set contains 1000 samples per class. All other experimental settings follow Subsec. 4.2.

**Results:** As shown in Fig. 7(a, b), MNIST outperformed CIFAR-10-Gray, which can be attributed to its near-zero variance (Fig. 7(d)). Direct encoding of RGB CIFAR-10 performed worse than grayscale CIFAR-10 due to additional encoding layers (Fig. 7(c)). As the number of repetitions increased, we observed that training accuracy increased while test accuracy decreased, leading to larger generalization error.

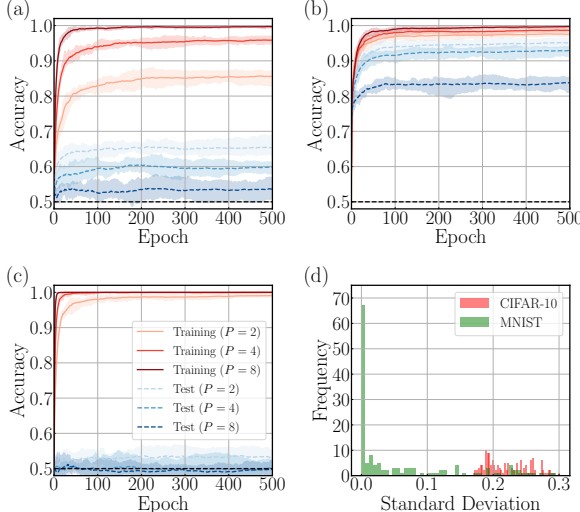

*Figure 7.* Results of real-world datasets: (a) Grayscale CIFAR-10, (b) MNIST, (c) RGB CIFAR-10, (d) Standard deviations of Grayscale CIFAR-10 and MNIST. Solid and dashed lines represent the mean training accuracy and test accuracy, respectively, across 10 independent runs with different random seeds. Shaded areas indicate the minimum and maximum values over these runs. Colors represent different number of repetitions ($P = 2, 4, 8$).

Additionally, we conducted regression experiments, analyzed the relationships among prediction error, training error, generalization error, and accuracy (App. G), and examined how increasing sample size and model complexity (repetition number) impacts these three types of errors.

## 5. Discussion

In this paper, we assume that the data follows an independent Gaussian distribution. While this assumption may appear strong given that real-world data elements are typically correlated, we emphasize several important points. First, our primary objective is to demonstrate the limitations of data re-uploading models with deep encoding layers. The fact that their predictive capability approaches random guessing under this assumption sufficiently illustrates this point. Sec-

ond, as validated in Sec. 4, our conclusions remain valid even when the data elements are not independent. Third, for certain tasks such as regression, data elements can be approximated as independent after feature engineering (such as principal component analysis).

We adopt the Gaussian distribution due to its prevalence and convenience for proof, but our conclusions are not limited to Gaussian distributions. The key property we require is that the distribution of data $x$ satisfies $\mathbb{E}[\cos(x)] = \gamma \cos(\mu)$, where $|\gamma| < 1$. For example, with a uniform distribution over $[\mu - a, \mu + a]$, we have $\mathbb{E}[\cos(x)] = \sin(a)/a \cdot \cos(\mu)$, where $|\gamma| = |\sin(a)/a| < 1$ when $a \neq 0$.

Finally, we note that our assumption implies each element provides independent information. When elements in high-dimensional data are strongly correlated, the limitations discussed in this paper may not occur. However, such data essentially reduces to one-dimensional information, for more details, see App. H.

## Impact Statement

Our paper aims to highlight the limitations of data re-uploading models with deep encoding layers in supervised quantum machine learning tasks. We provide theoretical guidance for the architectural design of data re-uploading models. Specifically, when using data re-uploading models to process high-dimensional data, employing a limited number of qubits with deep encoding layers is undesirable. Future work could focus on enhancing the predictive performance of large-scale data re-uploading models with shallow encoding layers.

## Acknowledgments

This work is supported by NSFC (Grant No. 62173201) and the Innovation Program for Quantum Science and Technology (Grants No. 2021ZD0300200).

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

# A. Terminology about data re-uploading

In this appendix, we clarify several key terminologies related to data re-uploading in this paper.

**Data re-uploading encoding:** The data re-uploading encoding discussed in this paper refers to an encoding paradigm that maps classical data into quantum circuits, characterized by two key properties:

- Insertion of parameterized quantum gates between encoding gates.

- Repeated uploading of data through identical encoding gates into the quantum circuit.

This encoding paradigm encompasses two special cases:

- When no parameterized quantum gates exist between encoding gates (only identity or fixed quantum gates), with parameterized gates appearing only after all encoding gates, the data re-uploading encoding becomes equivalent to angle encoding (encoding data through rotation gates) in the encoding-variational paradigm;

- When data is uploaded only once without repetition, which we consider as a special case of repeated uploading with repetition count equal to one.

When data re-uploading encoding satisfies both special cases simultaneously - that is, data is uploaded only once and only identity or fixed quantum gates (such as CNOT or CZ) exist between encoding gates - the data re-uploading encoding discussed in this paper reduces to the scenario discussed in the paper (Li et al., 2022).

**Data re-uploading circuits:** The quantum circuits that satisfy the data re-uploading encoding paradigm. Data re-uploading circuit using parameters $\boldsymbol{\theta}$ with respect to data $\boldsymbol{x}$ can be represented as a unitary operator $V(\boldsymbol{x}, \boldsymbol{\theta})$, which encodes data $\boldsymbol{x}$ into a quantum state $\rho(\boldsymbol{x}, \boldsymbol{\theta}) = V(\boldsymbol{x}, \boldsymbol{\theta})\rho_0 V(\boldsymbol{x}, \boldsymbol{\theta})^\dagger$, where $\rho_0$ is the initial state of the data re-uploading circuit.

**Data re-uploading models:** The machine learning models with data re-uploading circuits is data re-uploading models. For different machine learning tasks, the data re-uploading models will choose different observables to measure the encoded state $\rho(\boldsymbol{x}, \boldsymbol{\theta})$ generated by the data re-uploading circuits. The output of the data re-uploading model is the expectation value of the observable $H$ with respect to the encoded state, i.e., $h(\boldsymbol{x}, \boldsymbol{\theta}) = \mathrm{Tr}\,[H\rho(\boldsymbol{x}, \boldsymbol{\theta})]$. The relationship between the data re-uploading circuit and the data re-uploading model is shown in Fig. A.1.

*Figure A.1.* The data re-uploading circuit and data re-uploading model.

**Encoding Layer:** In a data re-uploading circuit, we define the number of encoding layers $L$ as the number of encoding gate layers used in one repetition cycle. Notably, in a data re-uploading circuit with $P$ repetitions, only $L$ distinct encoding layers are repeatedly used. The number of encoding layers in the circuit architecture corresponds to the number of data chunks, where each data chunk requires one encoding layer for uploading.

**Repetition:** In a data re-uploading circuit, the repetition count $P$ refers to the number of times the data is uploaded into the quantum circuit. While the same encoding layers are used across all $P$ repetitions, the parameterized gates inserted between encoding layers may have different structures and parameters.

**Depth:** In a data re-uploading circuit, if the structure of parameterized gates between any two encoding gates is identical (though parameters may differ), the circuit can be unfolded into $PL$ layers with identical structures but different parameters. The depth of such a circuit refers to the number of these layer structures, which equals $PL$. The relationship between encoding layer, repetition, and depth is shown in Fig. A.2.

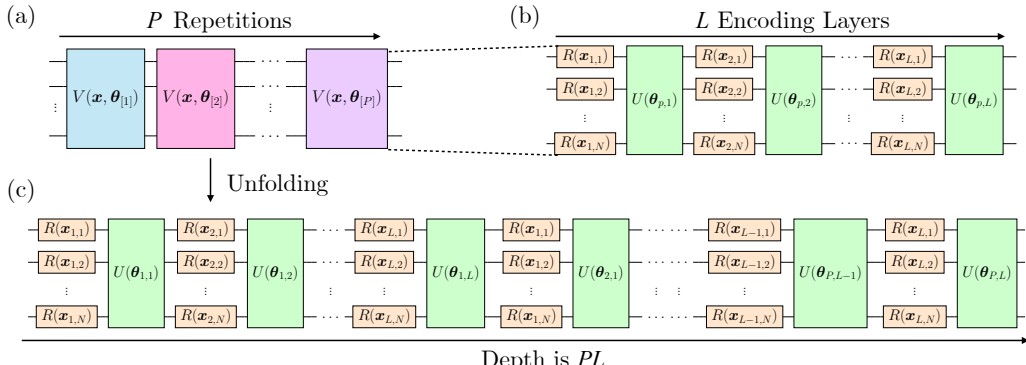

*Figure A.2.* (a) Illustration of Repetition, (b) Illustration of Encoding Layer, (c) Illustration of Depth.

## B. Measures of Quantum State Distinguishability

We introduce several commonly used measures of distinguishability for quantum states: trace distance, fidelity, affinity, and Petz-Rényi divergence. These measures play crucial roles in quantifying the distinguishability between quantum states. In particular, we establish the relationship between Petz-Rényi-2 divergence and trace distance, which serves as the foundation for our theoretical analysis in this paper.

**Definition B.1** (Quantum Trace Distance). The quantum trace distance between quantum states $\rho_1$ and $\rho_2$ is defined as:

$$T(\rho_1, \rho_2) \coloneqq \frac{1}{2}\|\rho_1 - \rho_2\|_1,$$

where $\|\cdot\|_1$ denotes the Schatten 1-norm. The quantum trace distance satisfies all properties of a mathematical distance: non-negativity, symmetry, and the triangle inequality. Furthermore, $T(\rho_1, \rho_2) \in [0, 1]$ with equality to 0 if and only if $\rho_1 = \rho_2$.

The trace distance provides an upper bound on the distinguishability of quantum states through measurements with respect to any observable $H$ with eigenvalues bounded in $[-1, 1]$. Specifically, the difference in expectation values between states $\rho_1$ and $\rho_2$ is bounded by the trace distance. This fundamental relationship follows directly from Hölder's inequality (Watrous, 2018):

$$|\text{Tr}[H\rho_1] - \text{Tr}[H\rho_2]| \leqslant \|H\|_\infty \|\rho_1 - \rho_2\|_1 = 2T(\rho_1, \rho_2),$$

where $\|\cdot\|_\infty$ represents the Schatten $\infty$-norm (spectral norm). The spectral norm constraint $\|H\|_\infty \leqslant 1$ naturally follows from the eigenvalues of $H$ being bounded in $[-1, 1]$.

**Definition B.2** (Quantum Fidelity). The quantum fidelity between quantum states $\rho_1$ and $\rho_2$ is defined as:

$$F(\rho_1, \rho_2) \coloneqq \text{Tr}\left[\sqrt{\sqrt{\rho_1}\rho_2\sqrt{\rho_1}}\right].$$

The quantum fidelity satisfies $F(\rho_1, \rho_2) \in [0, 1]$, with equality to 1 if and only if $\rho_1 = \rho_2$.

The Fuchs-van de Graaf inequality (Watrous, 2018) establishes that the trace distance and fidelity between quantum states $\rho_1$ and $\rho_2$ satisfy:

$$1 - F(\rho_1, \rho_2) \leqslant T(\rho_1, \rho_2) \leqslant \sqrt{1 - F(\rho_1, \rho_2)^2}. \tag{B.1}$$

**Definition B.3** (Quantum Affinity). The quantum affinity between quantum states $\rho_1$ and $\rho_2$ is defined as:

$$A(\rho_1, \rho_2) := \mathrm{Tr}\left[\sqrt{\rho_1}\sqrt{\rho_2}\right].$$

The quantum affinity satisfies $A(\rho_1, \rho_2) \in [0, 1]$, with equality to 1 if and only if $\rho_1 = \rho_2$.

Given that $F(\rho_1, \rho_2) = \|\sqrt{\rho_1}\sqrt{\rho_2}\|_1 = \mathrm{Tr}\left[|\sqrt{\rho_1}\sqrt{\rho_2}|\right]$, quantum affinity and fidelity are related through:

$$F(\rho_1, \rho_2)^2 = \mathrm{Tr}\left[\left|\sqrt{\rho_1}\sqrt{\rho_2}\right|\right]^2 \geqslant \left|\mathrm{Tr}\left[\sqrt{\rho_1}\sqrt{\rho_2}\right]\right|^2 = A(\rho_1, \rho_2)^2. \tag{B.2}$$

**Definition B.4** (Quantum Petz-Rényi-$\alpha$ Divergence). The quantum Petz-Rényi-$\alpha$ divergence between quantum states $\rho_1$ and $\rho_2$ is defined as:

$$D_\alpha(\rho_1\|\rho_2) := \frac{1}{\alpha - 1}\log_2\left(\mathrm{Tr}\left[\rho_1^\alpha \rho_2^{1-\alpha}\right]\right), \tag{B.3}$$

where $\alpha \in (0, 1) \cup (1, \infty)$, with the support condition $\mathrm{supp}(\rho_1) \subseteq \mathrm{supp}(\rho_2)$. The quantum Petz-Rényi-$\alpha$ divergence satisfies $D_\alpha(\rho_1\|\rho_2) \geqslant 0$, with equality if and only if $\rho_1 = \rho_2$.

**Definition B.5** (Quantum Relative Entropy). The quantum relative entropy between quantum states $\rho_1$ and $\rho_2$ is defined as:

$$S(\rho_1\|\rho_2) := \mathrm{Tr}[\rho_1(\log_2\rho_1 - \log_2\rho_2)],$$

where the support condition $\mathrm{supp}(\rho_1) \subseteq \mathrm{supp}(\rho_2)$ must be satisfied. The quantum relative entropy satisfies $S(\rho_1\|\rho_2) \geqslant 0$, with equality if and only if $\rho_1 = \rho_2$.

The quantum Petz-Rényi-$\alpha$ divergence has several important special cases that are worth highlighting.

- For $\alpha = 1$, the quantum Petz-Rényi-$\alpha$ divergence reduces to the quantum relative entropy:

$$D_1(\rho_1\|\rho_2) := \lim_{\alpha \to 1} D_\alpha(\rho_1\|\rho_2) = S(\rho_1\|\rho_2).$$

- For $\alpha = \frac{1}{2}$, it relates to quantum affinity:

$$D_{\frac{1}{2}}(\rho_1\|\rho_2) = -2\log_2\left(\mathrm{Tr}\left[\sqrt{\rho_1}\sqrt{\rho_2}\right]\right) = -2\log_2 A(\rho_1, \rho_2).$$

- For $\alpha = 2$, we obtain the divergence considered in this work:

$$D_2(\rho_1\|\rho_2) = \log_2\left(\mathrm{Tr}\left[\rho_1^2\rho_2^{-1}\right]\right). \tag{B.4}$$

As $\alpha \mapsto D_\alpha(\rho_1\|\rho_2)$ is monotonically increasing on $(0, +\infty)$ (Mosonyi & Hiai, 2011), we have:

$$-\log_2 A(\rho_1, \rho_2) = D_{\frac{1}{2}}(\rho_1\|\rho_2) \leqslant D_2(\rho_1\|\rho_2). \tag{B.5}$$

**Lemma B.6.** *For quantum states $\rho_1$ and $\rho_2$, their trace distance and Petz-Rényi-2 divergence satisfy:*

$$T(\rho_1, \rho_2) \leqslant \sqrt{1 - \frac{1}{2^{D_2(\rho_1\|\rho_2)}}}.$$

*Proof.* Combining the trace distance-fidelity inequality in Eq. (B.1) with the fidelity-affinity relation in Eq. (B.2), we obtain:

$$T(\rho_1, \rho_2) \leqslant \sqrt{1 - F(\rho_1, \rho_2)^2} \leqslant \sqrt{1 - A(\rho_1, \rho_2)^2}.$$

The result follows from the affinity-divergence inequality in Eq. (B.5). □

# C. Proof of Dependence on the Number of Encoding Layers

## C.1. Pauli Basis Decomposition

Let $Q_i$ and $P_i$ denote the Pauli basis elements before and after applying unitary gate $U$, respectively. The transformation of an $N$-qubit quantum state $\rho$ under $U$ can be expressed as:

$$
U\rho U^\dagger = \frac{1}{2^N} \left( \sum_{Q_i \in \{I,X,Y,Z\}^{\otimes N}} U(\alpha_i Q_i) U^\dagger \right)
$$

$$
= \frac{1}{2^N} \left( \sum_{P_i \in \{I,X,Y,Z\}^{\otimes N}} \beta_i P_i \right).
$$

Throughout the analysis, we adopt the Pauli basis ordering: $\{I, Z, X, Y\}$ for single-qubit systems and its $N$-fold tensor product $\{I, Z, X, Y\}^{\otimes N}$ for $N$-qubit systems.

Denoting $\boldsymbol{\alpha}$ and $\boldsymbol{\beta}$ as the vectors of Pauli basis coefficients before and after the unitary transformation respectively, there exists a Pauli basis coefficient transfer matrix $H$ such that $\boldsymbol{\beta} = H\boldsymbol{\alpha}$. The transfer matrix $H$ can be expressed as a collection of column vectors $H = \begin{bmatrix} \boldsymbol{h}_1 & \cdots & \boldsymbol{h}_{4^N} \end{bmatrix}$, where each $\boldsymbol{h}_j = \begin{bmatrix} h_{1j} & \cdots & h_{4^N j} \end{bmatrix}^\top$ is a $4^N$-dimensional column vector representing the transformation of the $j$-th original Pauli basis element. The transformed Pauli coefficients $\boldsymbol{\beta}$ are obtained through a linear combination of these column vectors: $\boldsymbol{\beta} = \sum_{j=1}^{4^N} \alpha_j \boldsymbol{h}_j$.

Each element $h_{ij}$ of the column vector $\boldsymbol{h}_j$ quantifies how the original Pauli basis element $Q_j$ contributes to the coefficient of the transformed basis element $P_i$ under the unitary operation $U$. This relationship is formally expressed as:

$$
U(\alpha_j Q_j) U^\dagger = \sum_{i=1}^{4^N} \alpha_j h_{ij} P_i. \tag{C.1}
$$

Next, we will first study the form of the transfer matrix for general quantum gates in Subsec. C.2. Then, in Subsec. C.3 and Subsec. C.4, we will analyze the properties of the transfer matrix for encoding gates under data expectation. Finally, we prove that the quantum divergence between the expected state of data re-uploading circuit and the maximally mixed state decays exponentially with the number of encoding layers $L$ in Subsec. C.5.

## C.2. General Properties of Quantum Gates in Pauli Basis

Firstly, we consider the general property of the Pauli basis coefficient transfer matrix $H$ determined by arbitrary quantum gates.

**Theorem C.1.** *For an $N$-qubit quantum gate $U$, the corresponding Pauli basis transfer matrix $H$ exhibits the following properties:*

*(1) The transfer matrix $H$ is orthogonal.*

*(2) The transfer matrix $H$ can be written as a block matrix in the following form:*

$$
H = \begin{bmatrix} 1 & \\ & \mathcal{H} \end{bmatrix},
$$

*where the top-left element is 1, and matrix $\mathcal{H}$ is also orthogonal.*

*Proof.* (1) According to Eq. (C.1), we have:

$$
U Q_j U^\dagger = \sum_{i=1}^{4^N} h_{ij} P_i.
$$

Taking the trace of the square of both sides and utilizing the properties of Pauli matrices $\mathrm{Tr}\left[P_i^2\right] = 2^N$ and $\mathrm{Tr}\left[P_i P_l\right] = 0$

for $P_i \neq P_l$, we obtain $\mathrm{Tr}\left[(UQ_jU^\dagger)^2\right] = 2^N$ for the left side, while the right side yields:

$$2^N = \mathrm{Tr}\left[\left(\sum_{i=1}^{4^N} h_{ij}P_i\right)^2\right] = \sum_{i=1}^{4^N} h_{ij}^2 \,\mathrm{Tr}\left[(P_i)^2\right] + \sum_{P_i,P_l\in\{I,X,Y,Z\}^{\otimes N},P_i\neq P_l} h_{ij}h_{lj}\,\mathrm{Tr}\left[P_iP_l\right]$$

$$= 2^N \sum_{i=1}^{4^N} h_{ij}^2 = 2^N \boldsymbol{h}_j^\top \boldsymbol{h}_j.$$

Therefore, $\boldsymbol{h}_i^\top \boldsymbol{h}_i = 1$.

Next, considering the action of quantum gate $U$ on another Pauli basis element $Q_k$ of the original qubits:

$$UQ_kU^\dagger = \sum_{l=1}^{4^N} h_{lk}P_l.$$

Then,

$$(UQ_jU^\dagger)(UQ_kU^\dagger) = \sum_{i=1}^{4^N} h_{ij}h_{ik}\,(P_i)^2 + \sum_{P_i,P_l\in\{I,X,Y,Z\}^{\otimes N},P_i\neq P_l} h_{ij}h_{lk}P_iP_l.$$

Taking the trace of both sides and using the properties of Pauli matrices $\mathrm{Tr}\left[P_i^2\right] = 2^N$ and $\mathrm{Tr}\left[P_iP_l\right] = 0$ for $P_i \neq P_l$, we obtain:

$$0 = \mathrm{Tr}\left[(UQ_jU^\dagger)(UQ_kU^\dagger)\right] = \sum_{i=1}^{4^N} h_{ij}h_{ik}\,\mathrm{Tr}\left[(P_i)^2\right] + \sum_{P_i,P_l\in\{I,X,Y,Z\}^{\otimes N},P_i\neq P_l} h_{ij}h_{lk}\,\mathrm{Tr}\left[P_iP_l\right]$$

$$= 2^N \sum_{i=1}^{4^N} h_{ij}h_{ik} = 2^N \boldsymbol{h}_j^\top \boldsymbol{h}_k.$$

This shows $\boldsymbol{h}_j^\top \boldsymbol{h}_k = \boldsymbol{h}_k^\top \boldsymbol{h}_j = 0$ for $j \neq k$. Thus,

$$H^\top H = HH^\top = I,$$

proving $H$ is orthogonal.

(2) Since the first element of the Pauli basis is $I^{\otimes N}$, the vector $\boldsymbol{h}_1$ in the first column of matrix $H$ represents the influence of the original identity operator $I^{\otimes N}$ on other Pauli basis elements after the action of gate $U$. From the unitary property $UU^\dagger = I$, we obtain:

$$\boldsymbol{h}_1 = \left[1, 0, \cdots, 0\right]^\top.$$

Now consider the first row elements $h_{1j}$ of matrix $H$. For the $j$-th Pauli basis element $Q_j$ (where $Q_j \neq I^{\otimes N}$) of the original quantum state, after the action of gate $U$:

$$UQ_jU^\dagger = h_{1j}I^{\otimes N} + \sum_{i=2}^{4^N} h_{ij}P_i.$$

Taking the trace of both sides and using the property of Pauli matrices $\mathrm{Tr}\left[P_i\right] = \mathrm{Tr}\left[Q_i\right] = 0, P_i \neq I^{\otimes N}, Q_i \neq I^{\otimes N}$, we obtain $h_{1j} = 0$.

Therefore, matrix $H$ can be written as:

$$H = \begin{bmatrix} 1 & \\ & \mathcal{H} \end{bmatrix},$$

and since $H$ is orthogonal, $\mathcal{H}$ is also orthogonal. $\qquad\square$

For the first property, the orthogonality of transfer matrix $H$ reflects an important physical principle: unitary operations preserve the purity of quantum states. When we express a pure state $\rho$ in the Pauli basis with coefficients $\boldsymbol{\alpha}$, its purity condition gives:

$$\text{Tr}[\rho^2] = \frac{1}{4^N} \left( \sum_{i=1}^{4^N} \alpha_i^2 \, \text{Tr}\left[P_i^2\right] + \sum_{P_i, P_j \in \{I, X, Y, Z\}^{\otimes N}, P_i \neq P_j} \alpha_i \alpha_j \, \text{Tr}\left[P_i P_j\right] \right)$$

$$= \frac{1}{2^N} \sum_{i=1}^{4^N} \alpha_i^2 = \frac{1}{2^N} \|\boldsymbol{\alpha}\|_2^2 = 1,$$

where we have used the Pauli basis properties that $\text{Tr}[P_i^2] = 2^N$ and $\text{Tr}[P_i P_j] = 0$ for $P_i \neq P_j$. Under unitary evolution, the transformed coefficients $\boldsymbol{\beta} = H\boldsymbol{\alpha}$ must maintain this unit norm, i.e., $\|\boldsymbol{\beta}\|_2^2 = \boldsymbol{\alpha}^\top H^\top H \boldsymbol{\alpha} = \|\boldsymbol{\alpha}\|_2^2$, which is guaranteed by $H$ being orthogonal.

The second property demonstrates that the transfer matrix preserves the fundamental quantum property of trace preservation. In the Pauli basis expansion, only the identity matrix $I^{\otimes N}$ contributes to the trace of a quantum state. Consequently, the coefficient associated with the identity term must remain invariant under unitary evolution. This fundamental constraint manifests in the block-diagonal structure of the transfer matrix $H$, where the identity component is decoupled from the other Pauli basis elements.

### C.3. Properties of Encoding Gate in Pauli Basis

We now analyze the transfer matrix of the encoding gate $R(\boldsymbol{x})$ defined in Eq. (1), where $R(\boldsymbol{x}) = R_z(x_3) R_y(x_2) R_z(x_1)$. Since single-qubit states can be decomposed in the Pauli basis $\{I, X, Y, Z\}$, the transfer matrices of $R_z(x)$ and $R_y(x)$ are determined by their action on this basis. We begin by establishing the fundamental commutation relations of Pauli matrices:

**Lemma C.2.** *Let $X, Y, Z$ be Pauli matrices, then:*

$$[Z, X] = 2\mathrm{i}Y, [Z, Y] = -2\mathrm{i}X;$$
$$[Y, Z] = 2\mathrm{i}X, [Y, X] = -2\mathrm{i}Z,$$

*where $[\cdot, \cdot]$ denotes the Lie bracket.*

The following lemma establishes the fundamental transformation rule of $R_P(x)$ gate acting on Pauli basis states.

**Lemma C.3.** *For single-qubit Pauli rotation gate $R_P(x)$ acting on single-qubit Pauli basis $Q$, where $P \in \{X, Y, Z\}$, $Q \in \{I, Z, X, Y\}$, we have:*

$$R_P(x) Q R_P(x)^\dagger = \begin{cases} Q & Q = I \text{ or } P = Q \\ \cos(x)Q - \sin(x)H & Q \neq I \text{ and } P \neq Q \end{cases},$$

*where $H = \mathrm{i}/2[P, Q]$, and $[\cdot, \cdot]$ denotes the Lie bracket.*

*Proof.* Since

$$R_P(x) = \cos\left(\frac{x}{2}\right) I - \mathrm{i}\sin\left(\frac{x}{2}\right) P,$$

we have

$$R_P(x) Q R_P^\dagger(x) = \left[\cos\left(\frac{x}{2}\right) I - \mathrm{i}\sin\left(\frac{x}{2}\right) P\right] Q \left[\cos\left(\frac{x}{2}\right) I + \mathrm{i}\sin\left(\frac{x}{2}\right) P\right]$$

$$= \cos^2\left(\frac{x}{2}\right) Q + \sin^2\left(\frac{x}{2}\right) PQP - \mathrm{i}\sin\left(\frac{x}{2}\right)\cos\left(\frac{x}{2}\right) PQ + \mathrm{i}\cos\left(\frac{x}{2}\right)\sin\left(\frac{x}{2}\right) QP.$$

By the properties of Pauli matrices, when $Q = I$ or $P = Q$, we have $PQP = Q$ and $PQ = QP$, therefore

$$R_P(x) Q R_P(x)^\dagger = \cos^2\left(\frac{x}{2}\right) Q + \sin^2\left(\frac{x}{2}\right) Q = Q$$

When $P \neq Q$ and $Q \neq I$, we have $PQP = -Q$, thus:

$$R_P(x) Q R_P(x)^\dagger = \left[\cos^2\left(\frac{x}{2}\right) - \sin^2\left(\frac{x}{2}\right)\right] Q - \left[\mathrm{i}\sin\left(\frac{x}{2}\right)\cos\left(\frac{x}{2}\right)\right] (PQ - QP)$$

$$= \cos(x)Q - \sin(x)H,$$

where $H = \mathrm{i}/2(PQ - QP)$. $\qquad\square$

Based on the fundamental commutation relations of Pauli matrices in Lemma C.2 and the transformation rules for single-qubit Pauli rotation gates in Lemma C.3, we can derive the transformation rules for $R_z(x)$ and $R_y(x)$ gates acting on Pauli basis states.

By Lemma C.3, for Pauli basis $Q = I$ or $Z$, we have $R_z(x)QR_z(x)^\dagger = Q$. When $Q = X$ or $Q = Y$, let $X^{\mathrm{old}}, Y^{\mathrm{old}}$ denote the Pauli basis of the original quantum state, and $X^{\mathrm{new}}, Y^{\mathrm{new}}$ denote the Pauli basis after the quantum gate $R_z(x)$ is applied. Combined with Lemma C.2, we have:

$$R_z(x)X^{\mathrm{old}}R_z(x)^\dagger = \cos(x)X^{\mathrm{new}} + \sin(x)Y^{\mathrm{new}},$$
$$R_z(x)Y^{\mathrm{old}}R_z(x)^\dagger = -\sin(x)X^{\mathrm{new}} + \cos(x)Y^{\mathrm{new}}.$$

Therefore, the transfer matrix of quantum gate $R_z(x)$ is:

$$T_z(x) = \begin{bmatrix} 1 & 0 & 0 & 0 \\ 0 & 1 & 0 & 0 \\ 0 & 0 & \cos(x) & -\sin(x) \\ 0 & 0 & \sin(x) & \cos(x) \end{bmatrix}. \tag{C.2}$$

Similarly, by Lemma C.3, for Pauli basis $Q = I$ or $Y$, we have $R_y(x)QR_y^\dagger(x) = Q$. When $Q = Z$ or $Q = X$, let $Z^{\mathrm{old}}, X^{\mathrm{old}}$ denote the Pauli basis of the original quantum state, and $Z^{\mathrm{new}}, X^{\mathrm{new}}$ denote the Pauli basis after the quantum gate $R_y(x)$ is applied. Combined with Lemma C.2, we have:

$$R_y(x)Z^{\mathrm{old}}R_y(x)^\dagger = \cos(x)Z^{\mathrm{new}} + \sin(x)X^{\mathrm{new}},$$
$$R_y(x)X^{\mathrm{old}}R_y(x)^\dagger = -\sin(x)Z^{\mathrm{new}} + \cos(x)X^{\mathrm{new}}.$$

Therefore, the transfer matrix of quantum gate $R_y(x)$ is:

$$T_y(x) = \begin{bmatrix} 1 & 0 & 0 & 0 \\ 0 & \cos(x) & -\sin(x) & 0 \\ 0 & \sin(x) & \cos(x) & 0 \\ 0 & 0 & 0 & 1 \end{bmatrix}. \tag{C.3}$$

### C.4. Properties of Expectation of Encoding Gate in Pauli Basis

The transfer matrix of $R_z(x)$ and $R_y(x)$ also satisfies the properties stated in Lemma C.1. However, under the expectation over data distribution, the transfer matrix of $R_z(x)$ and $R_y(x)$ are not orthogonal and cannot preserve the purity of quantum states.

Now, we introduce the following lemma to calculate the expectation of cosine and sine functions over Gaussian distribution.

**Lemma C.4.** *Let random variable* $x \sim \mathcal{N}(\mu, \sigma^2)$, *then*

$$\mathbb{E}_x[\cos(x)] = \mathrm{e}^{-\frac{\sigma^2}{2}}\cos(\mu); \quad \mathbb{E}_x[\sin(x)] = \mathrm{e}^{-\frac{\sigma^2}{2}}\sin(\mu).$$

Based on Lemma C.4 and transfer matrices of $R_z(x)$ and $R_y(x)$ in Eq. (C.2) and Eq. (C.3), we can get the expected transfer matrices of $R_z(x)$ and $R_y(x)$ under the data distribution.

**Lemma C.5.** *Assume the data* $x \sim \mathcal{N}(\mu, \sigma^2)$, *then the expected transfer matrices of* $R_z(x)$ *and* $R_y(x)$ *under the data*

*distribution are:*

$$\mathbb{E}_x[T_z(x)] = \begin{bmatrix} 1 & 0 & 0 & 0 \\ 0 & 1 & 0 & 0 \\ 0 & 0 & e^{-\frac{\sigma^2}{2}}\cos(\mu) & -e^{-\frac{\sigma^2}{2}}\sin(\mu) \\ 0 & 0 & e^{-\frac{\sigma^2}{2}}\sin(\mu) & e^{-\frac{\sigma^2}{2}}\cos(\mu) \end{bmatrix},$$

$$\mathbb{E}_x[T_y(x)] = \begin{bmatrix} 1 & 0 & 0 & 0 \\ 0 & e^{-\frac{\sigma^2}{2}}\cos(\mu) & -e^{-\frac{\sigma^2}{2}}\sin(\mu) & 0 \\ 0 & e^{-\frac{\sigma^2}{2}}\sin(\mu) & e^{-\frac{\sigma^2}{2}}\cos(\mu) & 0 \\ 0 & 0 & 0 & 1 \end{bmatrix}.$$

Therefore, we can get the expected transfer matrix of $R(\boldsymbol{x})$ under the data distribution.

**Lemma C.6.** *(From the work (Li et al., 2022)) Assume the data $\boldsymbol{x} = [x_1, x_2, x_3]$ follows independent Gaussian distributions, i.e., $x_i \sim \mathcal{N}(\mu_i, \sigma_i^2), i \in [3]$. The expected transfer matrix of encoding gate $R(\boldsymbol{x})$ under the data distribution is:*

$$\mathbb{E}_{\boldsymbol{x}}[T(\boldsymbol{x})] = \begin{bmatrix} 1 & 0 & 0 & 0 \\ 0 & t_{zz} & t_{xz} & t_{yz} \\ 0 & t_{zx} & t_{xx} & t_{yx} \\ 0 & t_{zy} & t_{xy} & t_{yy} \end{bmatrix},$$

*where*

$$t_{zz} = A_2 \cos(\mu_2),$$
$$t_{zx} = A_2 \sin(\mu_2) A_3 \cos(\mu_3),$$
$$t_{zy} = A_2 \sin(\mu_2) A_3 \sin(\mu_3),$$
$$t_{xz} = -A_2 \sin(\mu_2) A_1 \cos(\mu_1),$$
$$t_{xx} = A_2 \cos(\mu_2) A_1 \cos(\mu_1) A_3 \cos(\mu_3) - A_1 \sin(\mu_1) A_3 \sin(\mu_3),$$
$$t_{xy} = A_2 \cos(\mu_2) A_1 \cos(\mu_1) A_3 \sin(\mu_3) + A_1 \sin(\mu_1) A_3 \cos(\mu_3),$$
$$t_{yz} = A_2 \sin(\mu_2) A_1 \sin(\mu_1),$$
$$t_{yx} = -A_2 \cos(\mu_2) A_1 \sin(\mu_1) A_3 \cos(\mu_3) - A_1 \cos(\mu_1) A_3 \sin(\mu_3)$$
$$t_{yy} = -A_2 \cos(\mu_2) A_1 \sin(\mu_1) A_3 \sin(\mu_3) + A_1 \cos(\mu_1) A_3 \cos(\mu_3)$$

*where $A_i = e^{-\frac{\sigma_i^2}{2}}$.*

*Proof.* Since

$$\mathbb{E}_{\boldsymbol{x}}[T(\boldsymbol{x})] = \mathbb{E}_{\boldsymbol{x}}[T_z(x_3)T_y(x_2)T_z(x_1)] = \mathbb{E}_{x_3}[T_z(x_3)]\mathbb{E}_{x_2}[T_y(x_2)]\mathbb{E}_{x_1}[T_z(x_1)].$$

According to Lemma C.5, we have:

$$\mathbb{E}_{\boldsymbol{x}}[T(\boldsymbol{x})] = \begin{bmatrix} 1 & 0 & 0 & 0 \\ 0 & 1 & 0 & 0 \\ 0 & 0 & A_3\cos(\mu_3) & -A_3\sin(\mu_3) \\ 0 & 0 & A_3\sin(\mu_3) & A_3\cos(\mu_3) \end{bmatrix}\begin{bmatrix} 1 & 0 & 0 & 0 \\ 0 & A_2\cos(\mu_2) & -A_2\sin(\mu_2) & 0 \\ 0 & A_2\sin(\mu_2) & A_2\cos(\mu_2) & 0 \\ 0 & 0 & 0 & 1 \end{bmatrix}\begin{bmatrix} 1 & 0 & 0 & 0 \\ 0 & 1 & 0 & 0 \\ 0 & 0 & A_1\cos(\mu_1) & -A_1\sin(\mu_1) \\ 0 & 0 & A_1\sin(\mu_1) & A_1\cos(\mu_1) \end{bmatrix}.$$

$\square$

According to the Lemma C.1, the expected transfer matrix $\mathbb{E}_{\boldsymbol{x}}[T(\boldsymbol{x})]$ in Lemma C.6 can be written as a block matrix $\begin{bmatrix} 1 & \\ & \mathcal{T} \end{bmatrix}$, where

$$\mathcal{T} = \begin{bmatrix} t_{zz} & t_{xz} & t_{yz} \\ t_{zx} & t_{xx} & t_{yx} \\ t_{zy} & t_{xy} & t_{yy} \end{bmatrix}. \tag{C.4}$$

The properties of matrix $T$ mainly depend on the properties of matrix $\mathcal{T}$, so we will focus on analyzing matrix $\mathcal{T}$ next.

**Lemma C.7.** *(From the work (Li et al., 2022), Lemma S3) Given a Hermitian matrix $H \in \mathbb{C}^{n \times n}$ with all eigenvalues not exceeding $\lambda$, for any matrix $Q \in \mathbb{C}^{n \times n}$ with maximum singular value not exceeding $s$, the maximum eigenvalue of matrix $Q^\dagger H Q$ does not exceed $s^2\lambda$.*

**Lemma C.8.** *Assume the encoded data $\boldsymbol{x} = [x_1, x_2, x_3]$ follows independent Gaussian distributions, i.e., $x_i \sim \mathcal{N}(\mu_i, \sigma_i^2)$, with $\sigma_i^2 \geqslant \sigma^2, i \in [3]$. The expected transfer matrix of encoding gate $R(\boldsymbol{x}) = R_z(x_3)R_y(x_2)R_z(x_1)$ can be written as*
$$\mathbb{E}_{\boldsymbol{x}}[T(\boldsymbol{x})] = \begin{bmatrix} 1 & \\ & \mathcal{T} \end{bmatrix}, \text{ where the maximum eigenvalue of matrix } \mathcal{T}^\top \mathcal{T} \text{ does not exceed } e^{-\sigma^2}, \text{ and matrix } \mathcal{T}^\top \mathcal{T} \text{ is Hermitian.}$$

*Proof.* According to the proof of Lemma C.6, matrix $\mathcal{T}$ can be decomposed as:

$$\mathcal{T} = \begin{bmatrix} 1 & 0 & 0 \\ 0 & A_3 \cos(\mu_3) & -A_3 \sin(\mu_3) \\ 0 & A_3 \sin(\mu_3) & A_3 \cos(\mu_3) \end{bmatrix} \begin{bmatrix} A_2 \cos(\mu_2) & -A_2 \sin(\mu_2) & 0 \\ A_2 \sin(\mu_2) & A_2 \cos(\mu_2) & 0 \\ 0 & 0 & 1 \end{bmatrix} \begin{bmatrix} 1 & 0 & 0 \\ 0 & A_1 \cos(\mu_1) & -A_1 \sin(\mu_1) \\ 0 & A_1 \sin(\mu_1) & A_1 \cos(\mu_1) \end{bmatrix}$$

$$= \begin{bmatrix} 1 & 0 & 0 \\ 0 & \cos(\mu_3) & -\sin(\mu_3) \\ 0 & \sin(\mu_3) & \cos(\mu_3) \end{bmatrix} \begin{bmatrix} A_2 & 0 & 0 \\ 0 & A_2 A_3 & 0 \\ 0 & 0 & A_3 \end{bmatrix} \begin{bmatrix} \cos(\mu_2) & -\sin(\mu_2) & 0 \\ \sin(\mu_2) & \cos(\mu_2) & 0 \\ 0 & 0 & 1 \end{bmatrix} \begin{bmatrix} 1 & 0 & 0 \\ 0 & A_1 \cos(\mu_1) & -A_1 \sin(\mu_1) \\ 0 & A_1 \sin(\mu_1) & A_1 \cos(\mu_1) \end{bmatrix}.$$

Let $Q = \begin{bmatrix} 1 & 0 & 0 \\ 0 & A_1 \cos(\mu_1) & -A_1 \sin(\mu_1) \\ 0 & A_1 \sin(\mu_1) & A_1 \cos(\mu_1) \end{bmatrix}, R = \begin{bmatrix} \cos(\mu_2) & -\sin(\mu_2) & 0 \\ \sin(\mu_2) & \cos(\mu_2) & 0 \\ 0 & 0 & 1 \end{bmatrix}$, we have:

$$\mathcal{T}^\top \mathcal{T} = Q^\top R^\top \begin{bmatrix} A_2^2 & 0 & 0 \\ 0 & (A_2 A_3)^2 & 0 \\ 0 & 0 & A_3^2 \end{bmatrix} RQ.$$

Since $(\mathcal{T}^\top \mathcal{T})^\dagger = \mathcal{T}^\top \mathcal{T}$, matrix $\mathcal{T}^\top \mathcal{T}$ is Hermitian. As the maximum singular values of matrices $Q$ and $R$ do not exceed 1, and the maximum eigenvalue of matrix $\begin{bmatrix} A_2^2 & 0 & 0 \\ 0 & (A_2 A_3)^2 & 0 \\ 0 & 0 & A_3^2 \end{bmatrix}$ is $\max\{A_2^2, A_3^2\}$, by assumption, $A_2, A_3$ are both no greater than $e^{-\frac{\sigma^2}{2}}$. According to Lemma C.7, the maximum eigenvalue of matrix $\mathcal{T}^\top \mathcal{T}$ does not exceed $e^{-\sigma^2}$. $\square$

## C.5. Effect of Encoding Layers

The following theorem establishes that as the number of encoding layers $L$ increases, the expected encoded state approaches the maximally mixed state exponentially.

**Lemma C.9.** *(From the work (Li et al., 2022), Lemma S2) Given a Hermitian matrix $H \in \mathbb{C}^{n \times n}$ with all eigenvalues no greater than $\lambda$, and an $n$-dimensional vector $x$, we have*

$$\boldsymbol{x}^\dagger H \boldsymbol{x} \leq \|\boldsymbol{x}\|_2^2 \lambda,$$

*where $\| \cdot \|_2$ denotes the $l_2$-norm.*

**Theorem C.10** (Theorem 3.1 in the main text). *Consider an $N$-qubit data re-uploading circuit with $L$ encoding layers and without repetition ($P = 1$), which encodes data $\boldsymbol{x} \in \mathbb{R}^{3NL}$ into the circuit, where each data point follows an independent Gaussian distribution, i.e., $x_{l,n,i} \sim \mathcal{N}(\mu_{l,n,i}, \sigma_{l,n,i}^2)$ and $\sigma_{l,n,i}^2 \geqslant \sigma^2$. Let $\rho(\boldsymbol{x}, \boldsymbol{\theta})$ denote the $N$-qubit encoded state. Then the quantum divergence between the expected state $\mathbb{E}[\rho] = \mathbb{E}_{\boldsymbol{x}}[\rho(\boldsymbol{x}, \boldsymbol{\theta})]$ and the maximally mixed state $\rho_I = \frac{I}{2^N}$ satisfies:*

$$D_2 \left( \mathbb{E}[\rho] \| \rho_I \right) \leqslant \log_2 \left( 1 + (2^N - 1)e^{-L\sigma^2} \right).$$

*Proof.* Let $\boldsymbol{\alpha}$ be the Pauli basis vector of the initial state $\rho_0 = (|0\rangle \langle 0|)^{\otimes N}$ of the $N$-qubit data re-uploading circuit. Since $|0\rangle \langle 0| = \frac{1}{2}(I + Z)$, we have

$$\boldsymbol{\alpha} = \bigotimes_{n=1}^{N} \begin{bmatrix} 1 & 1 & 0 & 0 \end{bmatrix}.$$

Let $\boldsymbol{\beta}$ be the Pauli basis vector of the expected encoded state $\mathbb{E}_{\boldsymbol{x}}[\rho(\boldsymbol{x}, \boldsymbol{\theta})]$ with respect to data $\boldsymbol{x}$, i.e.,

$$\mathbb{E}[\rho] = \mathbb{E}_{\boldsymbol{x}}[\rho(\boldsymbol{x}, \boldsymbol{\theta})] = \frac{1}{2^N} \left( \sum_{P_i \in \{I, X, Y, Z\}^{\otimes N}} \beta_i P_i \right),$$

therefore,

$$(\mathbb{E}[\rho])^2 = \frac{1}{4^N}\left(\left(\sum_{i=1}^{4^n}\beta_i^2\right)P_i^2 + \sum_{P_i,P_j\in\{I,X,Y,Z\}^{\otimes N},P_i\neq P_j}\beta_i\beta_j P_i P_j\right).$$

Since $\mathrm{Tr}\left[P_i^2\right] = 2^N$ and for different Pauli matrices $P_i \neq P_j$, $\mathrm{Tr}\left[P_i P_j\right] = 0$, we have:

$$\mathrm{Tr}\left[(\mathbb{E}[\rho])^2\right] = \frac{1}{2^N}\sum_{i=1}^{4^n}\beta_i^2 = \frac{1}{2^N}\boldsymbol{\beta}^\top\boldsymbol{\beta}.$$

Let $H_l$ be the transfer matrix of the $l$-th layer parameterized gate, and $\mathbb{E}_{\boldsymbol{x}_{[l]}}[T(\boldsymbol{x}_{[l]})]$ be the expected transfer matrix of the $l$-th layer encoding gate with respect to data:

$$\mathbb{E}_{\boldsymbol{x}_{[l]}}[T(\boldsymbol{x}_{[l]})] = \bigotimes_{n=1}^{N}\mathbb{E}_{\boldsymbol{x}_{l,n}}[T(\boldsymbol{x}_{l,n})],$$

then we can obtain

$$\boldsymbol{\beta} = H_L\mathbb{E}_{\boldsymbol{x}_{[L]}}[T(\boldsymbol{x}_{[L]})]\cdots H_2\mathbb{E}_{\boldsymbol{x}_{[2]}}[T(\boldsymbol{x}_{[2]})]H_1\mathbb{E}_{\boldsymbol{x}_{[1]}}[T(\boldsymbol{x}_{[1]})]\boldsymbol{\alpha}.$$

Since the first element of $\boldsymbol{\alpha}$ corresponds to the coefficient of identity matrix $I^{\otimes N}$ which is 1, we have $\boldsymbol{\alpha} = \begin{bmatrix}1\\\boldsymbol{\alpha}^\circ\end{bmatrix}$. By Lemma C.1, $H_l, \mathbb{E}_{\boldsymbol{x}_{[l]}}[T(\boldsymbol{x}_{[l]})], l\in[L]$ can be written as block matrices:

$$H_l = \begin{bmatrix}1 & \\ & \mathcal{H}_l\end{bmatrix}; \mathbb{E}_{\boldsymbol{x}_{[l]}}[T(\boldsymbol{x}_{[l]})] = \begin{bmatrix}1 & \\ & \mathcal{T}_l\end{bmatrix},$$

where $\mathcal{H}_l$ is orthogonal, the maximum eigenvalue of $\mathcal{T}_l^\top\mathcal{T}_l$ does not exceed $e^{-\sigma^2}$, and $\mathcal{T}_l^\top\mathcal{T}_l$ is Hermitian, therefore

$$\boldsymbol{\beta}^\top\boldsymbol{\beta} = \begin{bmatrix}1 & (\boldsymbol{\alpha}^\circ)^\top\end{bmatrix}\begin{bmatrix}1 & \\ & \mathcal{T}_1^\top\mathcal{H}_1^\top\cdots\mathcal{T}_L^\top\mathcal{H}_L^\top\end{bmatrix}\begin{bmatrix}1 & \\ & \mathcal{H}_L\mathcal{T}_L\cdots\mathcal{H}_1\mathcal{T}_1\end{bmatrix}\begin{bmatrix}1\\\boldsymbol{\alpha}^\circ\end{bmatrix}$$
$$= 1 + (\boldsymbol{\alpha}^\circ)^\top\mathcal{T}_1^\top\mathcal{H}_1^\top\cdots\mathcal{T}_L^\top\ \mathcal{T}_L\cdots\mathcal{H}_1\mathcal{T}_1\boldsymbol{\alpha}^\circ.$$

By repeatedly using Lemma C.7, the maximum eigenvalue of matrix $\mathcal{T}_1^\top\mathcal{H}_1^\top\cdots\mathcal{T}_L^\top\ \mathcal{T}_L\cdots\mathcal{H}_1\mathcal{T}_1$ does not exceed $e^{-L\sigma^2}$, and $\boldsymbol{\alpha}^\circ$ has $2^N - 1$ elements equal to 1, so $\|\boldsymbol{\alpha}^\circ\| = 2^N - 1$. According to Lemma C.9, we have:

$$\boldsymbol{\beta}^\top\boldsymbol{\beta} \leqslant 1 + (2^N - 1)e^{-L\sigma^2}.$$

Therefore,

$$D_2(\mathbb{E}[\rho]||\rho_I) = \log_2\left(\mathrm{Tr}\left[\mathbb{E}[\rho]^2\cdot\left(\frac{I}{2^N}\right)^{-1}\right]\right) = \log_2\left(2^N\cdot\mathrm{Tr}\left[\mathbb{E}[\rho]^2\right]\right)$$
$$= \log_2(\boldsymbol{\beta}^\top\boldsymbol{\beta}) \leqslant \log_2\left(1 + (2^N - 1)e^{-L\sigma^2}\right).$$

$$\square$$

**Corollary C.11.** *Consider an $N$-qubit data re-uploading circuit with $L$ encoding layers and without repetition ($P = 1$), where the encoded data follows the independent Gaussian distribution defined in Theorem 3.1. Let $h_1(\boldsymbol{x},\boldsymbol{\theta})$ be its output with respect to an observable $H$ whose eigenvalues lie in $[-1, 1]$. When $L \geqslant \frac{1}{\sigma^2}[(N + 2)\ln 2 + 2\ln(\frac{1}{\epsilon})]$, we have*

$$\left|\mathbb{E}_{\boldsymbol{x}}[h_1(\boldsymbol{x},\boldsymbol{\theta})] - h_I\right| \leqslant \epsilon,$$

*where $h_I = \mathrm{Tr}\left[H\rho_I\right]$, and $\rho_I = \frac{I}{2^N}$ is the maximally mixed state of $N$ qubits.*

*Proof.* Let $\mathbb{E}[\rho] = \mathbb{E}_{\boldsymbol{x}}[\rho(\boldsymbol{x}, \boldsymbol{\theta})]$. By Theorem C.10, when $L \geqslant \frac{1}{\sigma^2}[(N+2)\ln 2 + 2\ln(\frac{1}{\epsilon})]$, the divergence between $\mathbb{E}[\rho]$ and the maximally mixed state $\rho_I$ satisfies:

$$D_2(\mathbb{E}[\rho]||\rho_I) \leqslant 1 + (2^N - 1)e^{-L\sigma^2} \leqslant 1 + \frac{(2^N - 1)\epsilon^2}{2^{N+2}}$$
$$\leqslant 1 + \frac{2^N \cdot \epsilon^2}{4 \cdot 2^N} \leqslant \frac{4 + \epsilon^2}{4}.$$

By the relationship between trace distance and divergence given in Lemma B.6, we have

$$T(\mathbb{E}[\rho], \rho_I) \leqslant \sqrt{1 - \frac{4}{4 + \epsilon^2}} = \frac{\epsilon}{\sqrt{4 + \epsilon^2}} \leqslant \frac{\epsilon}{2}.$$

Therefore, by Hölder's inequality (Watrous, 2018), we have:

$$|\mathrm{Tr}\,[H\mathbb{E}[\rho]] - \mathrm{Tr}\,[H\rho_I]| \leqslant \|H\|_\infty \|\mathbb{E}[\rho] - \rho_I\|_1 \leqslant 2T(\mathbb{E}[\rho], \rho_I) \leqslant \epsilon,$$

where $\|H\|_\infty$ is the Schatten-$\infty$ norm (spectral norm) of $H$. $\qquad\square$

# D. Proof of Dependence on the Number of Repetitions

## D.1. Periodicity of Encoding Gates

Consider the encoding gate defined in Eq. (1) which consists of rotation gates $R_z(x)$ and $R_y(x)$. For any Pauli operator $R_P(\boldsymbol{x}), P \in \{Z, Y\}$ acting on quantum state $\rho$, we can derive:

$$R_P(x)\rho R_P(x)^\dagger$$
$$= \cos^2\left(\frac{x}{2}\right)\rho + \sin^2\left(\frac{x}{2}\right)P\rho P - \mathrm{i}\sin\left(\frac{x}{2}\right)\cos\left(\frac{x}{2}\right)P\rho + \mathrm{i}\cos\left(\frac{x}{2}\right)\sin\left(\frac{x}{2}\right)\rho P$$
$$= \frac{1 + \cos(x)}{2}\rho + \frac{1 - \cos(x)}{2}P\rho P - \frac{1}{2}\mathrm{i}\sin(x)P\rho + \frac{1}{2}\mathrm{i}\sin(x)\rho P.$$

From this expression, it is evident that the mapping $x \mapsto R_P(x)\rho R_P(x)^\dagger, P \in \{Z, Y\}$ exhibits periodicity with period $2\pi$.

## D.2. Approximating Circuit

For an approximate data $\tilde{x}_d = \sum_{j=-3}^q b_j 2^{-j}$, where $b_j \in \{0, 1\}$, we can construct a controlled rotation gate $\mathrm{C} - R_z(2^{-j})$ with auxiliary qubits $|b_j\rangle$ as controls and working qubits as targets. This controlled rotation gate approximates the rotation gate in working qubits as:

$$R_z\left(\sum_{j=-3}^q b_j 2^{-j}\right) = R_z(x_d - \epsilon_q) = R_z(\tilde{x}_d),$$

Similarly, for the $R_y$ gate, we can approximate it using controlled rotation gates $\mathrm{C} - R_y(2^{-j})$ with auxiliary qubits $|b_j\rangle$ as controls and working qubits as targets. This controlled rotation gate approximates the rotation gate in working qubits as:

$$R_y\left(\sum_{j=-3}^q b_j 2^{-j}\right) = R_y(x_d - \epsilon_q) = R_y(\tilde{x}_d),$$

where $\epsilon_q = x_d - \tilde{x}_d$.

In the original data re-uploading circuits shown in Fig. 2, the chunk of data in the $l$-th encoding layer is $\boldsymbol{x}_{[l]} = [\boldsymbol{x}_{l,1}, \cdots, \boldsymbol{x}_{l,N}]^\top$. Here, $\tilde{\boldsymbol{x}}_{[l]}$ represents the $(q+3)$-bit binary approximation of each data component in $\boldsymbol{x}_{[l]}$. The quantum state corresponding to binary string $\tilde{\boldsymbol{x}}_{[l]}$ in the approximating circuit is $|\tilde{\boldsymbol{x}}_{[l]}\rangle\langle\tilde{\boldsymbol{x}}_{[l]}|$, and the initial state of approximating circuit is:

$$\tilde{\rho}_0(\boldsymbol{x}) = \rho_{w,0} \otimes \left(\bigotimes_{l=1}^L |\tilde{\boldsymbol{x}}_{[l]}\rangle\langle\tilde{\boldsymbol{x}}_{[l]}|\right),$$

where $\rho_{w,0}$ is the initial state on working qubits, typically $|0\rangle\langle0|$. Since each encoding layer in the data re-uploading circuit uses quantum gates defined in Eq. (1) for data encoding, the data-independent controlled gates used for any $\tilde{\boldsymbol{x}}_{[l]}$ are identical. Denoting the controlled gate as $CR$ and the quantum gate applied in the approximating circuit during the $p$-th repetition as $V_p$, we have:

$$V_p\left(\boldsymbol{\theta}_{[p]}\right) = \prod_{l=1}^{\overleftarrow{L}} U_l\left(\boldsymbol{\theta}_{p,l}\right) CR,$$

where $\boldsymbol{\theta}_{[p]} = [\boldsymbol{\theta}_{p,1}, \cdots, \boldsymbol{\theta}_{p,L}]$. After $P$ repetitions, the quantum state of approximating circuit is:

$$\tilde{\rho}_P(\boldsymbol{x}, \boldsymbol{\theta}) = V_P\left(\boldsymbol{\theta}_{[P]}\right) \cdots V_1\left(\boldsymbol{\theta}_{[1]}\right) \tilde{\rho}_0\left(\boldsymbol{x}\right) V_1\left(\boldsymbol{\theta}_{[1]}\right)^\dagger \cdots V_P\left(\boldsymbol{\theta}_{[P]}\right)^\dagger,$$

and the corresponding quantum state on working qubits is:

$$\tilde{\rho}_{w,P}(\boldsymbol{x}, \boldsymbol{\theta}) = \mathrm{Tr}_{\boldsymbol{x}}\left[V_P\left(\boldsymbol{\theta}_{[P]}\right) \cdots V_1\left(\boldsymbol{\theta}_{[1]}\right) \tilde{\rho}_0\left(\boldsymbol{x}\right) V_1\left(\boldsymbol{\theta}_{[1]}\right)^\dagger \cdots V_P\left(\boldsymbol{\theta}_{[P]}\right)^\dagger\right],$$

where $\mathrm{Tr}_{\boldsymbol{x}}[\cdot]$ denotes the partial trace over the auxiliary qubits that contained the information of data $\boldsymbol{x}$. $\tilde{\rho}_{w,P}(\boldsymbol{x}, \boldsymbol{\theta})$ is equivalent to the quantum state:

$$\rho_P(\tilde{\boldsymbol{x}}, \boldsymbol{\theta}) = \left(\prod_{p=1}^{\overleftarrow{P}} \prod_{l=1}^{\overleftarrow{L}} U_l\left(\boldsymbol{\theta}_{p,l}\right) R_l\left(\tilde{\boldsymbol{x}}_{[l]}\right)\right) \rho_0 \left(\prod_{p=1}^{P} \prod_{l=1}^{L} R_l\left(\tilde{\boldsymbol{x}}_{[l]}\right)^\dagger U_l\left(\boldsymbol{\theta}_{p,l}\right)^\dagger\right),$$

where $\rho_0 = \rho_{w,0} = |0\rangle\langle0|$. Therefore, for any observable $H$ acting only on working qubits, we have:

$$\mathrm{Tr}\left[H\tilde{\rho}_P(\boldsymbol{x}, \boldsymbol{\theta})\right] = \mathrm{Tr}\left[H\tilde{\rho}_{w,P}(\boldsymbol{x}, \boldsymbol{\theta})\right] = \mathrm{Tr}\left[H\rho_P(\tilde{\boldsymbol{x}}, \boldsymbol{\theta})\right]. \tag{D.1}$$

### D.3. Approximation Error Analysis

To investigate the approximation error between the approximating circuit in Fig. 3 and data re-uploading circuit in Fig. 2, we give the following definitions and lemmas:

**Definition D.1** (Diamond Distance). For quantum channels $\mathcal{N}_{\mathcal{A}\to\mathcal{B}}$ and $\mathcal{M}_{\mathcal{A}\to\mathcal{B}}$, their diamond distance is defined as:

$$\|\mathcal{N}_{A\to B} - \mathcal{M}_{A\to B}\|_\diamond := \sup_{\rho_{RA} \in \mathcal{D}(\mathcal{H}_{RA})} \|(\mathcal{I}_R \otimes \mathcal{N}_{A\to B})(\rho_{RA}) - (\mathcal{I}_R \otimes \mathcal{M}_{A\to B})(\rho_{RA})\|_1,$$

where $\|\cdot\|_1$ is the Schatten-1 norm.

**Lemma D.2** (From the work (Watrous, 2018), Proposition 3.48). *For any completely positive and trace-preserving maps $\mathcal{A}, \mathcal{B}, \mathcal{C}, \mathcal{D}$, where $\mathcal{B}$ and $\mathcal{D}$ map from $n$-qubit systems to $m$-qubit systems, and $\mathcal{A}$ and $\mathcal{C}$ map from $m$-qubit systems to $k$-qubit systems, the following inequality holds:*

$$\|\mathcal{AB} - \mathcal{CD}\|_\diamond \leqslant \|\mathcal{A} - \mathcal{C}\|_\diamond + \|\mathcal{B} - \mathcal{D}\|_\diamond.$$

**Lemma D.3** (From the work(Caro et al., 2022), Lemma B.5). *Let $\mathcal{U}(\rho) = U\rho U^\dagger$, $\mathcal{V}(\rho) = V\rho V^\dagger$ be unitary channels, then*

$$\frac{1}{2}\|\mathcal{U} - \mathcal{V}\|_\diamond \leqslant \|U - V\|_\infty,$$

*where $\|\cdot\|_\infty$ is the Schatten-$\infty$ norm (spectral norm).*

**Lemma D.4.** *Given any Pauli matrix $P \in \{X, Y, Z\}$ and two parameters $\phi_1, \phi_2 \in [0, 2\pi]$, construct two rotation operators $R(\phi_1) = e^{-i\frac{\phi_1}{2}P}, R(\phi_2) = e^{-i\frac{\phi_2}{2}P}$, then the Schatten-$\infty$ norm between these two rotation operators satisfies:*

$$\|R(\phi_1) - R(\phi_2)\|_\infty \leqslant \frac{1}{2}|\phi_1 - \phi_2|.$$

*Proof.* By the definition of rotation operators, we have:

$$R(\phi_1) - R(\phi_2) = \left[\cos\left(\frac{\phi_1}{2}\right) - \cos\left(\frac{\phi_2}{2}\right)\right] I - \mathrm{i}\left[\sin\left(\frac{\phi_1}{2}\right) - \sin\left(\frac{\phi_2}{2}\right)\right] P.$$

The singular values of this matrix are $2|\sin(\frac{\phi_1-\phi_2}{4})|$ with multiplicity $2^n$. Using the inequality $|\sin(x)| \leq |x|$ for all real $x$, we obtain:

$$\|R(\phi_1) - R(\phi_2)\|_\infty = 2\sin\left(\frac{\phi_1-\phi_2}{4}\right) \leqslant \frac{1}{2}|\phi_1 - \phi_2|.$$

This completes the proof. $\qquad\square$

The following theorem illustrates the relationship between the approximation error of the approximating circuit and the number of binary approximation bits used for the data components.

**Theorem D.5.** *Consider an $N$-qubit data re-uploading circuit with $L$ encoding layers and $P$ repetitions. Let $h_P(\boldsymbol{x}, \boldsymbol{\theta})$ be its output with respect to an observable $H$ whose eigenvalues lie in $[-1, 1]$. There exists an approximating circuit as shown in Fig. 3 with output $\tilde{h}_P(\boldsymbol{x}, \boldsymbol{\theta})$ with respect to $H$ on working qubits. When the number of approximation qubits $q$ used for each data satisfies $q \geqslant \lceil \log_2(3PLN/\delta) \rceil$, we have:*

$$\left| h_P(\boldsymbol{x}, \boldsymbol{\theta}) - \tilde{h}_P(\boldsymbol{x}, \boldsymbol{\theta}) \right| \leqslant \delta.$$

*Proof.* Define the quantum channel of unitary gates as $\mathcal{U}_{p,l}(\rho) = U_l(\boldsymbol{\theta}_{p,l})\rho U_l(\boldsymbol{\theta}_{p,l})^\dagger$. For an $N$-qubit circuit with encoding layers $L$, define the channel of a single data encoding rotation gate as $\mathcal{R}_P(x_{l,n,i})(\rho) = R_P(x_{l,n,i})\rho R_P(x_{l,n,i})^\dagger$, where $P = Y, Z$. The encoding channel for the $n$-th qubit at layer $l$ is $\mathcal{R}_{l,n}(\boldsymbol{x}_{l,n}) = \mathcal{R}_z(x_{l,n,3})\mathcal{R}_y(x_{l,n,2})\mathcal{R}_z(l, n, 1)$, and the encoding channel for layer $l$ is $\mathcal{R}_l(\boldsymbol{x}_{[l]}) = \prod_{n=1}^{N}\mathcal{R}_n(\boldsymbol{x}_{l,n})$.

Let the channel of the data re-uploading circuit be $\mathcal{E} = \overleftarrow{\prod}_{p=1}^{P} \overleftarrow{\prod}_{l=1}^{L} \mathcal{U}_{p,l}\mathcal{R}_l(\boldsymbol{x}_{[l]})$, and the channel of the approximating circuit be $\tilde{\mathcal{E}} = \overleftarrow{\prod}_{p=1}^{P} \overleftarrow{\prod}_{l=1}^{L} \mathcal{U}_{p,l}\mathcal{R}_l(\tilde{\boldsymbol{x}}_{[l]})$. Then $h_P(\boldsymbol{x}, \boldsymbol{\theta}) = \mathrm{Tr}\left[H\mathcal{E}(\rho_0)\right]$, $\tilde{h}_P(\boldsymbol{x}, \boldsymbol{\theta}) = \mathrm{Tr}\left[H\tilde{\mathcal{E}}(\rho_0)\right]$, therefore

$$
\begin{aligned}
\left| h_P(\boldsymbol{x}, \boldsymbol{\theta}) - \tilde{h}_P(\boldsymbol{x}, \boldsymbol{\theta}) \right| &= \left| \mathrm{Tr}\left[H\mathcal{E}(\rho_0)\right] - \mathrm{Tr}\left[H\tilde{\mathcal{E}}(\rho_0)\right] \right| \\
&\leqslant \|H\|_\infty \cdot \|\mathcal{E} - \tilde{\mathcal{E}}\|_\diamond && \text{(D.2)} \\
&\leqslant \sum_{p=1}^{P}\sum_{l=1}^{L} \|\mathcal{U}_{p,l} - \mathcal{U}_{p,l}\|_\diamond + P\sum_{l=1}^{L} \|\mathcal{R}_l(\boldsymbol{x}_{[l]}) - \mathcal{R}_l(\tilde{\boldsymbol{x}}_{[l]})\|_\diamond && \text{(D.3)} \\
&\leqslant P\sum_{l=1}^{L}\sum_{n=1}^{N} \|\mathcal{R}_{l,n}(\boldsymbol{x}_{l,n}) - \mathcal{R}_{l,n}(\tilde{\boldsymbol{x}}_{l,n})\|_\diamond && \text{(D.4)} \\
&\leqslant P\sum_{l=1}^{L}\sum_{n=1}^{N}\sum_{i=1}^{3} \|\mathcal{R}_{l,n,i}(x_{l,n,i}) - \mathcal{R}_{l,n,i}(\tilde{x}_{l,n,i})\|_\diamond && \text{(D.5)} \\
&\leqslant 2P\sum_{l=1}^{L}\sum_{n=1}^{N}\sum_{i=1}^{3} \|R_{l,n,i}(x_{l,n,i}) - R_{l,n,i}(\tilde{x}_{l,n,i})\|_\infty && \text{(D.6)} \\
&\leqslant P\sum_{l=1}^{L}\sum_{n=1}^{N}\sum_{i=1}^{3} |x_{l,n,i} - \tilde{x}_{l,n,i}| && \text{(D.7)} \\
&\leqslant 3NLP|\epsilon_q|, && \text{(D.8)}
\end{aligned}
$$

where Eq. (D.2) follows from Hölder's inequality, Eqs. (D.3), (D.4), (D.5) follow from Lemma D.2, Eq. (D.6) follows from Lemma D.3, and Eq. (D.7) follows from Lemma D.4. Therefore, when $|\epsilon_q| \leqslant \frac{\delta}{3PLN}$, i.e., $q \geqslant \lceil \log_2(\frac{3PLN}{\delta}) \rceil$, we have:

$$\left| h_P(\boldsymbol{x}, \boldsymbol{\theta}) - \tilde{h}_P(\boldsymbol{x}, \boldsymbol{\theta}) \right| \leqslant \delta.$$

$\qquad\square$

## D.4. Effect of Repeated Data Uploading

Next, we will prove that repeated data uploading cannot mitigate the limitations in predictive performance imposed by encoding layers.

**Theorem D.6** (Theorem 3.2 in the main text). *Consider an $N$-qubit data re-uploading circuit with $L$ encoding layers and $P$ repetitions, which encodes data $\boldsymbol{x} \in \mathbb{R}^{3NL}$ into the circuit, where each data point follows an independent Gaussian distribution, i.e., $x_{l,n,i} \sim \mathcal{N}(\mu_{l,n,i}, \sigma^2_{l,n,i})$ and $\sigma^2_{l,n,i} \geqslant \sigma^2$. Let $h_P(\boldsymbol{x}, \boldsymbol{\theta})$ be its output with respect to an observable $H$ whose eigenvalues lie in $[-1, 1]$. When $L \geqslant \frac{1}{\sigma^2}[(N+2)\ln 2 + 2\ln(\frac{1}{\epsilon})]$, we have*

$$\left|\mathbb{E}_{\boldsymbol{x}}[h_P(\boldsymbol{x}, \boldsymbol{\theta})] - h_I\right| \leqslant \epsilon,$$

*where $h_I = \mathrm{Tr}\,[H\rho_I]$ and $\rho_I = \frac{I}{2^N}$ is the $N$-qubit maximally mixed state.*

*Proof.* Let $\boldsymbol{\theta}_{[p]}$ denote the parameters used in the $p$-th repetition of the data re-uploading circuit, with $V_p(\boldsymbol{\theta}_{[p]})$ being the corresponding quantum gate in the approximating circuit used to approximate the quantum gate in the $p$-th repetition of the data re-uploading circuit.

We define $\boldsymbol{\theta}_{[1:P]}$ as the complete set of parameters across all $P$ repetitions, and $\tilde{\rho}_P(\boldsymbol{x}, \boldsymbol{\theta}_{[1:P]})$ as the final quantum state of the approximating circuit after $P$ repetitions of data re-uploading. Let $\tilde{\rho}_1(\boldsymbol{x}, \boldsymbol{\theta}_{[1]})$ be the quantum state of approximating circuit corresponding to data re-uploading without repetition ($P = 1$). Their measurement results with respect to an observable $H$ on working qubits are:

$$\tilde{h}_1(\boldsymbol{x}, \boldsymbol{\theta}) = \mathrm{Tr}\,\left[HV_1(\boldsymbol{\theta}_{[1]})\tilde{\rho}_0(\boldsymbol{x})V_1(\boldsymbol{\theta}_{[1]})^\dagger\right]$$
$$= \mathrm{Tr}\,\left[H\tilde{\rho}_1(\boldsymbol{x}, \boldsymbol{\theta}_{[1]})\right],$$
$$\tilde{h}_P(\boldsymbol{x}, \boldsymbol{\theta}) = \mathrm{Tr}\,\left[HV_P(\boldsymbol{\theta}_{[P]})\cdots V_1(\boldsymbol{\theta}_{[1]})\tilde{\rho}_0(\boldsymbol{x})V_1(\boldsymbol{\theta}_{[1]})^\dagger \cdots V_P(\boldsymbol{\theta}_{[P]})^\dagger\right]$$
$$= \mathrm{Tr}\,\left[H\tilde{\rho}_P(\boldsymbol{x}, \boldsymbol{\theta}_{[1:P]})\right].$$

Therefore,

$$\tilde{h}_P(\boldsymbol{x}, \boldsymbol{\theta}) = \mathrm{Tr}\,\left[HV_P(\boldsymbol{\theta}_{[P]})\cdots V_2(\boldsymbol{\theta}_{[2]})\tilde{\rho}_1(\boldsymbol{x}, \boldsymbol{\theta}_{[1]})V_2(\boldsymbol{\theta}_{[2]})^\dagger \cdots V_P(\boldsymbol{\theta}_{[P]})^\dagger\right]$$
$$= \mathrm{Tr}\,\left[V_2(\boldsymbol{\theta}_{[2]})^\dagger \cdots V_P(\boldsymbol{\theta}_{[P]})^\dagger HV_P(\boldsymbol{\theta}_{[P]})\cdots V_2(\boldsymbol{\theta}_{[2]})\tilde{\rho}_1(\boldsymbol{x}, \boldsymbol{\theta}_{[1]})\right]$$
$$= \mathrm{Tr}\,\left[H'(\boldsymbol{\theta}_{[2:P]})\tilde{\rho}_1(\boldsymbol{x}, \boldsymbol{\theta}_{[1]})\right],$$

where

$$H'(\boldsymbol{\theta}_{[2:P]}) = V_2(\boldsymbol{\theta}_{[2]})^\dagger \cdots V_P(\boldsymbol{\theta}_{[P]})^\dagger HV_P(\boldsymbol{\theta}_{[P]})\cdots V_2(\boldsymbol{\theta}_{[2]}).$$

Since $V(\boldsymbol{\theta}_{[p]})$ are all unitary matrices, $H'(\boldsymbol{\theta}_{[2:P]})$ remains an observable with eigenvalues in $[-1, 1]$.

Let the outputs of the original data re-uploading circuit without repetition and its approximating circuit with respect to the new observable $H' := H'(\boldsymbol{\theta}_{[2:P]})$ be:

$$h_1(\boldsymbol{x}, \boldsymbol{\theta}) = \mathrm{Tr}\,\left[H'\rho_1(\boldsymbol{x}, \boldsymbol{\theta}_{[1]})\right],$$
$$\tilde{h}'_1(\boldsymbol{x}, \boldsymbol{\theta}) = \mathrm{Tr}\,\left[H'\tilde{\rho}_1(\boldsymbol{x}, \boldsymbol{\theta}_{[1]})\right].$$

The output of the new observable $H'$ with respect to the $N$-qubit maximally mixed state $\rho_I = \frac{I}{2^N}$ is:

$$h'_I = \mathrm{Tr}\,[H'\rho_I] = \mathrm{Tr}\,\left[V_2(\boldsymbol{\theta}_{[2]})^\dagger \cdots V_P(\boldsymbol{\theta}_{[P]})^\dagger HV_P(\boldsymbol{\theta}_{[P]})\cdots V_2(\boldsymbol{\theta}_{[2]})\rho_I\right]$$
$$= \mathrm{Tr}\,\left[HV_P(\boldsymbol{\theta}_{[P]})\cdots V_2(\boldsymbol{\theta}_{[2]})\rho_I V_2(\boldsymbol{\theta}_{[2]})^\dagger \cdots V_P(\boldsymbol{\theta}_{[P]})^\dagger\right]$$
$$= \mathrm{Tr}\,[H\rho_I] = h_I.$$

Therefore, combining Theorem D.5, when $q \geqslant \left\lceil\log_2(\frac{3LN}{\delta})\right\rceil$, $|h_1(\boldsymbol{x}, \boldsymbol{\theta})| - \tilde{h}'_1(\boldsymbol{x}, \boldsymbol{\theta})| \leqslant \delta$, and according to Corollary C.11, when $L \geqslant \frac{1}{\sigma^2}[(N+2)\ln 2 + 2\ln(\frac{1}{\epsilon})]$, we have $|\mathbb{E}_{\boldsymbol{x}}[h'_1(\boldsymbol{x}, \boldsymbol{\theta})] - h_I| \leqslant \epsilon$, so

$$\left|\mathbb{E}_{\boldsymbol{x}}[\tilde{h}'_1(\boldsymbol{x}, \boldsymbol{\theta})] - h_I\right| \leqslant \left|\mathbb{E}_{\boldsymbol{x}}[\tilde{h}'_1(\boldsymbol{x}, \boldsymbol{\theta})] - \mathbb{E}_{\boldsymbol{x}}[h'_1(\boldsymbol{x}, \boldsymbol{\theta})]\right| + \left|\mathbb{E}_{\boldsymbol{x}}[h'_1(\boldsymbol{x}, \boldsymbol{\theta})] - h_I\right|$$
$$\leqslant \epsilon + \delta,$$

and

$$\mathbb{E}_{\boldsymbol{x}}[\tilde{h}_1'(\boldsymbol{x}, \boldsymbol{\theta})] = \mathbb{E}_{\boldsymbol{x}}\{\text{Tr}\,[H'\tilde{\rho}_1(\boldsymbol{x}, \boldsymbol{\theta})]\}$$
$$= \mathbb{E}_{\boldsymbol{x}}\left\{\text{Tr}\left[HV_P(\boldsymbol{\theta}_{[P]})\cdots V_1(\boldsymbol{\theta}_{[1]})\tilde{\rho}_0(\boldsymbol{x})V_1(\boldsymbol{\theta}_{[1]})^\dagger\cdots V_P(\boldsymbol{\theta}_{[P]})^\dagger\right]\right\}$$
$$= \mathbb{E}_{\boldsymbol{x}}[\tilde{h}_P(\boldsymbol{x}, \boldsymbol{\theta})].$$

Therefore, $|\mathbb{E}_{\boldsymbol{x}}[\tilde{h}_P(\boldsymbol{x}, \boldsymbol{\theta})] - h_I| \leqslant \epsilon + \delta$, and according to Theorem D.5, $|h_P(\boldsymbol{x}, \boldsymbol{\theta}) - \tilde{h}_P(\boldsymbol{x}, \boldsymbol{\theta})| \leqslant \delta$, so

$$\left|\mathbb{E}_{\boldsymbol{x}}[h_P(\boldsymbol{x}, \boldsymbol{\theta})] - h_I\right| \leqslant \epsilon + 2\delta.$$

Since $\delta$ can be arbitrarily small as $q \to \infty$, we have $|\mathbb{E}_{\boldsymbol{x}}[h_P(\boldsymbol{x}, \boldsymbol{\theta})] - h_I| \leqslant \epsilon$. $\qquad\square$

# E. Proof of Bounds on the Prediction Error

## E.1. Classification Problems

For binary classification problems, consider a training set $S$ consisting of samples $(\boldsymbol{x}, y)$ where $\boldsymbol{x} \sim \mathcal{D}_\mathcal{X}$ and class labels $\mathcal{Y} = \{0, 1\}$. The hypothesis $h_S$ generated by the data re-uploading model trained on dataset $S$ is defined as:

$$h_S(\boldsymbol{x}) = \text{Tr}\left[H_{y(\boldsymbol{x})}V(\boldsymbol{x}, \boldsymbol{\theta}^*)\rho_0 V(\boldsymbol{x}, \boldsymbol{\theta}^*)^\dagger\right],$$

where $\boldsymbol{\theta}^*$ are the parameters chosen during training, for $y(\boldsymbol{x}) \in \{0, 1\}$, $H_0 = |0\rangle\langle 0|$, $H_1 = |1\rangle\langle 1|$. $h_S(\boldsymbol{x}) \in [0, 1]$ represents the probability that the model predicts feature $\boldsymbol{x}$ belongs to the correct class. The prediction error is defined as:

$$R^C(h_S) = \mathbb{E}_{\boldsymbol{x}\sim\mathcal{D}_\mathcal{X}}[|1 - h_S(\boldsymbol{x})|]. \tag{E.1}$$

**Proposition E.1** (Proposition 3.3 in the main text). *Consider binary classification tasks where the input features $\boldsymbol{x}$ are drawn from distribution $\mathcal{D}_\mathcal{X}$ and the class labels belong to $\mathcal{Y} = \{0, 1\}$. Using the data re-uploading model with corresponding observables $H_0 = |0\rangle\langle 0|$ and $H_1 = |1\rangle\langle 1|$, the hypothesis generated by the data re-uploading model trained on dataset $S$ is given by $h_S(\boldsymbol{x}) = \text{Tr}\left[H_{y(\boldsymbol{x})}V(\boldsymbol{x}, \boldsymbol{\theta}^*)\rho_0 V(\boldsymbol{x}, \boldsymbol{\theta}^*)^\dagger\right]$. Here, $\boldsymbol{\theta}^*$ represents the parameters chosen during training, and $y(\boldsymbol{x})$ denotes the label associated with feature $\boldsymbol{x}$. The binary classification prediction error is defined in Eq. (E.1). When the expectation of $h_S(\boldsymbol{x})$ over $\mathcal{D}_\mathcal{X}$ satisfies $|\mathbb{E}_{\boldsymbol{x}\sim\mathcal{D}_\mathcal{X}}[h_S(\boldsymbol{x})] - h_I| \leqslant \epsilon$, the prediction error of this hypothesis under distribution $\mathcal{D}_\mathcal{X}$ is bounded by:*

$$\left|R^C(h_S) - \frac{1}{2}\right| \leqslant \epsilon.$$

*Proof.* Since for $H_0 = |0\rangle\langle 0|$, $H_1 = |1\rangle\langle 1|$, we have $h_I = \text{Tr}\left[H_{y(\boldsymbol{x})}\rho_I\right] = \frac{1}{2}$, therefore

$$\left|R^C(h_S) - \frac{1}{2}\right| = \left|\mathbb{E}_{\boldsymbol{x}\sim\mathcal{D}_\mathcal{X}}[1 - h_S(\boldsymbol{x})] - \frac{1}{2}\right|$$
$$= \left|\frac{1}{2} - \mathbb{E}_{\boldsymbol{x}\sim\mathcal{D}_\mathcal{X}}[h_S(\boldsymbol{x})]\right|$$
$$= \left|h_I - \mathbb{E}_{\boldsymbol{x}\sim\mathcal{D}_\mathcal{X}}[h_S(\boldsymbol{x})]\right| \leqslant \epsilon.$$

$\qquad\square$

## E.2. Regression Problems

For multidimensional functions $f(\boldsymbol{x})$ where $\boldsymbol{x} \sim \mathcal{D}_\mathcal{X}$ and function values are bounded in $[-1, 1]$, we consider the $N$-qubit data re-uploading model with the observable $H_L = \bigotimes_{n=1}^N Z_n$. The corresponding hypothesis function is:

$$h_S(\boldsymbol{x}) = \text{Tr}\left[H_L V(\boldsymbol{x}, \boldsymbol{\theta}^*)\rho_0 V^\dagger(\boldsymbol{x}, \boldsymbol{\theta}^*)\right],$$

where $\boldsymbol{\theta}^*$ are parameters chosen during training. We consider the prediction error:

$$R^L(h_S) = \mathop{\mathbb{E}}_{\boldsymbol{x} \sim \mathcal{D}_\mathcal{X}} [|f(\boldsymbol{x}) - h_S(\boldsymbol{x})|]. \tag{E.2}$$

This prediction error measures the difference between true labels $f(\boldsymbol{x})$ and model outputs $h_S(\boldsymbol{x})$.

**Proposition E.2** (Proposition 3.3 in the main text). *Consider regression tasks where the input features $\boldsymbol{x}$ are drawn from distribution $\mathcal{D}_\mathcal{X}$ and the function values are bounded in $[-1, 1]$. Using the data re-uploading model with the observable $H_L = \bigotimes_{n=1}^N Z_n$ having eigenvalues in $[-1, 1]$, the hypothesis generated by the data re-uploading model trained on dataset $S$ is $h_S(\boldsymbol{x}) = \mathrm{Tr}\left[H_L V(\boldsymbol{x}, \boldsymbol{\theta}^*) \rho_0 V^\dagger(\boldsymbol{x}, \boldsymbol{\theta}^*)\right]$, where $\boldsymbol{\theta}^*$ are parameters chosen during training. The regression prediction error is defined in Eq. (E.2). When the expectation of $h_S(\boldsymbol{x})$ satisfies $|\mathbb{E}_{\boldsymbol{x} \sim \mathcal{D}_\mathcal{X}}[h_S(\boldsymbol{x})] - h_I| \leqslant \epsilon$, the prediction error of this hypothesis under distribution $\mathcal{D}_\mathcal{X}$ satisfies:*

$$R^L(h_S) \geqslant \left| \mathop{\mathbb{E}}_{\boldsymbol{x} \sim \mathcal{D}_\mathcal{X}} [f(\boldsymbol{x})] \right| - \epsilon.$$

*Proof.* Since for the observable $H_L = \bigotimes_{n=1}^N Z_n$, we have $h_I = \mathrm{Tr}\left[H_L \rho_I\right] = 0$, therefore

$$\begin{aligned}
R^L(h_S) &= \mathop{\mathbb{E}}_{\boldsymbol{x} \sim \mathcal{D}_\mathcal{X}} [|h_S(\boldsymbol{x}) - f(\boldsymbol{x})|] \\
&\geqslant \left| \mathop{\mathbb{E}}_{\boldsymbol{x} \sim \mathcal{D}_\mathcal{X}} [h_S(\boldsymbol{x})] - \mathop{\mathbb{E}}_{\boldsymbol{x} \sim \mathcal{D}_\mathcal{X}} [f(\boldsymbol{x})] \right| \\
&\geqslant \left| \mathop{\mathbb{E}}_{\boldsymbol{x} \sim \mathcal{D}_\mathcal{X}} [f(\boldsymbol{x})] \right| - \epsilon.
\end{aligned}$$

$\square$

# F. Proof of Difference Between Average and Expected Performance

In this appendix, we will explain the reason why we can use average performance as a surrogate of the expected performance.

## F.1. Useful Lemma

**Lemma F.1.** *For any $N$-qubit quantum state $\rho$, the following inequality holds:*

$$\frac{1}{2^N} \leqslant \mathrm{Tr}\left[\rho^2\right] \leqslant 1.$$

*Proof.* Firstly, According to the definition of purity, $\mathrm{Tr}\left[\rho^2\right] \leqslant 1$ is obvious.

Then, let $\lambda_1, \cdots, \lambda_{2^N}$ be the $2^N$ eigenvalues (with multiplicity) of the $N$-qubit quantum state. All quantum states satisfy the constraint $\sum_{i=1}^{2^N} \lambda_i = 1$. According to the AM-GM inequality:

$$\mathrm{Tr}\left[\rho^2\right] = \sum_{i=1}^{2^N} \lambda_i^2 \geqslant 2^N \cdot \sqrt[2^N]{\lambda_1 \cdots \lambda_{2^N}}.$$

The equality holds if and only if $\lambda_1 = \cdots = \lambda_{2^N}$, which corresponds to the $N$-qubit maximally mixed state where $\lambda_i = \frac{1}{2^N}, i \in [2^N]$, and $\mathrm{Tr}\left[\rho^2\right] = 2^N \cdot (1/2^N)^2 = 1/2^N$. $\square$

As the definition of Petz-Rényi-2 divergence in Eq. (B.4), for any $N$-qubit quantum state $\rho$, we have:

$$D_2(\rho \| \rho_I) = \log_2\left(\mathrm{Tr}\left[\rho^2 \rho_I^{-1}\right]\right) = \log_2\left(2^N \mathrm{Tr}\left[\rho^2\right]\right) \leqslant N.$$

Thus, for $N$-qubit quantum states, the Petz-Rényi-2 divergence between any quantum state and the maximally mixed state is upper bounded by $N$.

### F.2. Difference Between Average and Expected Petz-Rényi-2 Divergence

The following lemma tells us that in order to make the Petz-Rényi-2 divergence between the average quantum state and maximally mixed state close to the Petz-Rényi-2 divergence between the expected quantum state and maximally mixed state, we need to make the number of features $M$ large enough.

**Lemma F.2.** *Consider a dataset $S = \{(\boldsymbol{x}^{(m)}, y^{(m)})\}_{m=1}^{M}$ where the features $\boldsymbol{x}^{(m)}$ are independently sampled from a distribution $\mathcal{D}_{\mathcal{X}}$. Let $\rho(\boldsymbol{x}, \boldsymbol{\theta})$ represent the pure quantum state encoded by a data re-uploading circuit with parameters $\boldsymbol{\theta}$, where the feature $\boldsymbol{x}$ is encoded into the quantum state, and $\boldsymbol{\theta}$ are independent of the features in $S$. The average state is given by $\overline{\rho}_M := \frac{1}{M} \sum_{m=1}^{M} \rho(\boldsymbol{x}^{(m)}, \boldsymbol{\theta})$, while the expected state is $\mathbb{E}[\rho] := \mathbb{E}_{\boldsymbol{x} \sim \mathcal{D}_{\mathcal{X}}}[\rho(\boldsymbol{x}, \boldsymbol{\theta})]$. For any $\epsilon \in (0, 1)$, as $M \to \infty$, we have:*

$$|D_2(\overline{\rho}_M || \rho_I) - D_2(\mathbb{E}[\rho] || \rho_I)| \leqslant \epsilon,$$

*almost surely.*

*Proof.* The Petz-Rényi-2 divergence between quantum state $\rho$ and the maximally mixed state $\rho_I = \frac{I}{2^N}$ is:

$$D_2(\rho || \rho_I) = \log_2\left(2^N \operatorname{Tr}\left[\rho^2\right]\right) = N + \log_2\left(\operatorname{Tr}\left[\rho^2\right]\right). \tag{F.1}$$

Therefore,

$$
\begin{aligned}
|D_2(\overline{\rho}_M || \rho_I) - D_2(\mathbb{E}[\rho] || \rho_I)| &= \left|\log_2\left(\operatorname{Tr}\left[\overline{\rho}_M^2\right]\right) - \log_2\left(\operatorname{Tr}\left[\mathbb{E}[\rho]^2\right]\right)\right| \\
&= \left|\log_2\left(\frac{\operatorname{Tr}\left[\overline{\rho}_M^2\right]}{\operatorname{Tr}\left[\mathbb{E}[\rho]^2\right]}\right)\right|.
\end{aligned}
\tag{F.2}
$$

For notational simplicity, we omit the parameter $\boldsymbol{\theta}$ in the quantum states, i.e., $\rho(\boldsymbol{x}) = \rho(\boldsymbol{x}, \boldsymbol{\theta})$. Let $f(\boldsymbol{x}^{(1)}, \cdots, \boldsymbol{x}^{(n)}, \cdots, \boldsymbol{x}^{(M)}) = \operatorname{Tr}\left[\overline{\rho}_M^2\right] = \frac{1}{M^2} \sum_{m,n=1}^{M} \operatorname{Tr}\left[\rho(\boldsymbol{x}^{(m)})\rho(\boldsymbol{x}^{(n)})\right]$. For any $n \in [M]$, when replacing $\boldsymbol{x}^{(n)}$ with $(\boldsymbol{x}^{(n)})'$ and its corresponding quantum state $\rho'(\boldsymbol{x}^{(n)}) = \rho((\boldsymbol{x}^{(n)})')$, we have:

$$
\begin{aligned}
&\left|f(\boldsymbol{x}^{(1)}, \cdots, \boldsymbol{x}^{(n)}, \cdots, \boldsymbol{x}^{(M)}) - f(\boldsymbol{x}^{(1)}, \cdots, (\boldsymbol{x}^{(n)})', \cdots, \boldsymbol{x}^{(M)})\right| \\
&= \left|\frac{1}{M^2}\left[\sum_{m=1}^{M}\left(\operatorname{Tr}\left[\rho(\boldsymbol{x}^{(m)}\rho(\boldsymbol{x}^{(n)}))\right] - \operatorname{Tr}\left[\rho(\boldsymbol{x}^{(m)})\rho'(\boldsymbol{x}^{(n)})\right]\right)\right]\right| \\
&= \left|\frac{1}{M^2} \sum_{m=1}^{M} \operatorname{Tr}\left[\rho(\boldsymbol{x}^{(m)})\left\{\rho(\boldsymbol{x}^{(n)}) - \rho'(\boldsymbol{x}^{(n)})\right\}\right]\right| \\
&\leqslant \frac{1}{M^2} \sum_{m=1}^{M} \left\|\rho(\boldsymbol{x}^{(m)})\right\|_{\infty} \cdot \left\|\rho(\boldsymbol{x}^{(n)}) - \rho'(\boldsymbol{x}^{(n)})\right\|_1 \\
&\leqslant \frac{2}{M},
\end{aligned}
\tag{F.3}
$$

where $\|\cdot\|_{\infty}$ and $\|\cdot\|_1$ are the Schatten $\infty$-norm and Schatten 1-norm, respectively, and Eq. (F.3) is due to the Hölder's inequality (Watrous, 2018).

Let $\boldsymbol{X} = (\boldsymbol{x}^{(1)}, \cdots, \boldsymbol{x}^{(n)}, \cdots, \boldsymbol{x}^{(M)})$, by McDiarmid's inequality (Mohri, 2018), we have:

$$\mathop{\mathbb{P}}_{\boldsymbol{X} \sim \mathcal{D}_{\mathcal{X}}^{M}}\left(\left|f(\boldsymbol{x}^{(1)}, \cdots, \boldsymbol{x}^{(n)}, \cdots, \boldsymbol{x}^{(M)}) - \mathop{\mathbb{E}}_{\boldsymbol{X} \sim \mathcal{D}_{\mathcal{X}}^{M}}[f(\boldsymbol{x}^{(1)}, \cdots, \boldsymbol{x}^{(n)}, \cdots, \boldsymbol{x}^{(M)})]\right| \geqslant t\right) \leqslant 2\exp\left(\frac{-Mt^2}{2}\right).$$

Therefore,

$$\left|\operatorname{Tr}\left[\overline{\rho}_M^2\right] - \mathop{\mathbb{E}}_{\boldsymbol{X} \sim \mathcal{D}_{\mathcal{X}}^{M}}\left\{\operatorname{Tr}\left[\overline{\rho}_M^2\right]\right\}\right| \leqslant \epsilon \tag{F.4}$$

holds with probability at least $1 - 2e^{-M\epsilon^2/2}$.

We can expand the trace of the squared average state as follows:

$$\mathrm{Tr}\left[\overline{\rho}_M^2\right] = \frac{1}{M^2}\sum_{m,n=1}^{M}\mathrm{Tr}\left[\rho(\boldsymbol{x}^{(m)})\rho(\boldsymbol{x}^{(n)})\right]$$

$$= \frac{1}{M^2}\left(\sum_{m=1}^{M}\mathrm{Tr}\left[\rho(\boldsymbol{x}^{(m)})^2\right] + \sum_{m\neq n}\mathrm{Tr}\left[\rho(\boldsymbol{x}^{(m)})\rho(\boldsymbol{x}^{(n)})\right]\right)$$

$$= \frac{1}{M} + \frac{1}{M^2}\sum_{m\neq n}\mathrm{Tr}\left[\rho(\boldsymbol{x}^{(m)})\rho(\boldsymbol{x}^{(n)})\right],$$

where we have used the fact that each $\rho(\boldsymbol{x}^{(m)})$ is a pure state, implying $\rho(\boldsymbol{x}^{(m)})^2 = \rho(\boldsymbol{x}^{(m)})$ and $\mathrm{Tr}[\rho(\boldsymbol{x}^{(m)})^2] = 1$ for each $m$.

Given the independence of $\boldsymbol{x}^{(m)}$ and $\boldsymbol{x}^{(n)}$, we can derive the expectation of the trace of the squared average state as:

$$\mathop{\mathbb{E}}_{\boldsymbol{X}\sim\mathcal{D}_{\mathcal{X}^M}}\left\{\mathrm{Tr}\left[\overline{\rho}_M^2\right]\right\} = \frac{1}{M} + \frac{1}{M^2}\sum_{m\neq n}\mathrm{Tr}\left[\mathop{\mathbb{E}}_{\boldsymbol{X}\sim\mathcal{D}_{\mathcal{X}}^M}[\rho(\boldsymbol{x}^{(m)})\rho(\boldsymbol{x}^{(n)})]\right]$$

$$= \frac{1}{M} + \frac{1}{M^2}\sum_{m\neq n}\mathrm{Tr}\left[\mathop{\mathbb{E}}_{\boldsymbol{x}^{(m)}\sim\mathcal{D}_{\mathcal{X}}}[\rho(\boldsymbol{x}^{(m)})]\mathop{\mathbb{E}}_{\boldsymbol{x}^{(n)}\sim\mathcal{D}_{\mathcal{X}}}[\rho(\boldsymbol{x}^{(n)})]\right]$$

$$= \frac{1}{M} + \frac{1}{M^2}\sum_{m\neq n}\mathrm{Tr}\left[\mathbb{E}[\rho]^2\right]$$

$$= \frac{1}{M} + \frac{1}{M^2}(M^2 - M)\mathrm{Tr}\left[\mathbb{E}[\rho]^2\right]$$

$$= \frac{1}{M} + \left(1 - \frac{1}{M}\right)\mathrm{Tr}\left[\mathbb{E}[\rho]^2\right].$$

This leads to the following expression for the difference between the average state and expected state:

$$\left|\mathrm{Tr}\left[\overline{\rho}_M^2\right] - \mathop{\mathbb{E}}_{\boldsymbol{X}\sim\mathcal{D}_{\mathcal{X}^m}}\left\{\mathrm{Tr}\left[\overline{\rho}_M^2\right]\right\}\right| = \left|\mathrm{Tr}\left[\overline{\rho}_M^2\right] - \frac{1}{M} - \left(1 - \frac{1}{M}\right)\mathrm{Tr}\left[\mathbb{E}[\rho]^2\right]\right|.$$

Since $\mathrm{Tr}\left[\mathbb{E}[\rho]^2\right]$ is positive, According to Eq. (F.4), with probability at least $1 - 2e^{-M\epsilon^2/2}$ we have:

$$\left|\frac{\mathrm{Tr}\left[\overline{\rho}_M^2\right]}{\mathrm{Tr}\left[\mathbb{E}[\rho]^2\right]} - 1 - \frac{1}{M}\left(\frac{1}{\mathrm{Tr}\left[\mathbb{E}[\rho]^2\right]} - 1\right)\right| \leqslant \epsilon.$$

Then, applying Lemma F.1, with the same probability bound, we have:

$$\left|\frac{\mathrm{Tr}\left[\overline{\rho}_M^2\right]}{\mathrm{Tr}\left[\mathbb{E}[\rho]^2\right]} - 1\right| \leqslant \epsilon + \frac{1}{M}\left|\frac{1}{\mathrm{Tr}\left[\mathbb{E}[\rho]^2\right]} - 1\right| \leqslant \epsilon + \frac{2^N - 1}{M}.$$

We analyze the difference in Petz-Rényi-2 divergences in two cases based on the ratio $\mathrm{Tr}\left[\overline{\rho}_M^2\right]/\mathrm{Tr}\left[\mathbb{E}[\rho]^2\right]$.

Case 1: When the ratio is greater than or equal to 1, Eq. (F.2) implies that with probability at least $1 - 2e^{-M\epsilon^2/2}$:

$$|D_2(\overline{\rho}_M||\rho_I) - D_2(\mathbb{E}[\rho]||\rho_I)| = \log_2\left(\frac{\mathrm{Tr}\left[\overline{\rho}_M^2\right]}{\mathrm{Tr}\left[\mathbb{E}[\rho]^2\right]}\right) \leqslant \log_2\left(1 + \epsilon + \frac{2^N - 1}{M}\right). \tag{F.5}$$

Case 2: When the ratio is between 0 and 1, and $1 - \epsilon - \frac{2^{N-1}}{M} > 0$, we have with the same probability:

$$|D_2(\overline{\rho}_M||\rho_I) - D_2(\mathbb{E}[\rho]||\rho_I)| = -\log_2\left(\frac{\mathrm{Tr}\left[\overline{\rho}_M^2\right]}{\mathrm{Tr}\left[\mathbb{E}[\rho]^2\right]}\right) \leqslant -\log_2\left(1 - \epsilon - \frac{2^N - 1}{M}\right). \tag{F.6}$$

Asymptotically, as $M \to \infty$ with fixed $N$, based on Eq. (F.5) and Eq. (F.6), the difference in Petz-Rényi-2 divergences converges as:

$$|D_2(\overline{\rho}_M||\rho_I) - D_2(\mathbb{E}[\rho]||\rho_I)| \leqslant \max\{\log(1 + \epsilon), -\log(1 - \epsilon)\} \leqslant \epsilon + o(\epsilon).$$

Note that when $M \to \infty$, $1 - \epsilon - \frac{2^N - 1}{M} > 0$ is always satisfied. $\qquad\square$

### F.3. Difference Between Average and Expected Model Output

The following lemma shows that we can approximate the output of an observable $H$ on the expected quantum state by using the output of H on the average quantum state.

**Lemma F.3.** *Consider a dataset $S = \{(\boldsymbol{x}^{(m)}, y^{(m)})\}_{m=1}^M$ where features $\boldsymbol{x}^{(m)}$ are independently sampled from a distribution $\mathcal{D}_\mathcal{X}$. Let $\rho(\boldsymbol{x}, \boldsymbol{\theta})$ represent the pure quantum state encoded by a data re-uploading circuit with parameters $\boldsymbol{\theta}$, where the feature $\boldsymbol{x}$ is encoded into the quantum state, and $\boldsymbol{\theta}$ are independent of the features in S. We define the average state as $\overline{\rho}_M := \frac{1}{M} \sum_{m=1}^M \rho(\boldsymbol{x}^{(m)}, \boldsymbol{\theta})$ and the expected state as $\mathbb{E}[\rho] := \mathbb{E}_{\boldsymbol{x} \sim \mathcal{D}_\mathcal{X}}[\rho(\boldsymbol{x}, \boldsymbol{\theta})]$. For any observable H with eigenvalues bounded in $[-1, 1]$ and any $\epsilon \in (0, 1)$, the following inequality holds:*

$$\left| \mathrm{Tr}\,[H\overline{\rho}_M] - \mathrm{Tr}\,[H\mathbb{E}[\rho]] \right| \leqslant \epsilon$$

*holds with probability at least $1 - 2e^{-M\epsilon^2/2}$.*

*Proof.* For notational simplicity, we write $\rho(\boldsymbol{x}) = \rho(\boldsymbol{x}, \boldsymbol{\theta})$, omitting the explicit dependence on parameters $\boldsymbol{\theta}$. Let $\boldsymbol{X} = (\boldsymbol{x}^{(1)}, \cdots, \boldsymbol{x}^{(M)})$ where each feature vector is independently sampled from $\mathcal{D}_\mathcal{X}$. Since the parameters $\boldsymbol{\theta}$ are independent of the features, the sequence $\{\frac{1}{M} \mathrm{Tr}[H\rho(\boldsymbol{x}^{(m)})]\}_{m=1}^M$ forms a set of independent and identically distributed random variables. Furthermore, as the eigenvalues of $H$ lie in $[-1, 1]$, each term satisfies $-\frac{1}{M} \leqslant \frac{1}{M} \mathrm{Tr}[H\rho(\boldsymbol{x}^{(m)})] \leqslant \frac{1}{M}$ for all $m \in [M]$. Applying Hoeffding's inequality (Mohri, 2018), we obtain:

$$\mathbb{P}_{\boldsymbol{X} \sim \mathcal{D}_\mathcal{X}^M} \left( \left| \sum_{m=1}^M \frac{1}{M} \mathrm{Tr}\left[H\rho\left(\boldsymbol{x}^{(m)}\right)\right] - \mathbb{E}_{\boldsymbol{x} \sim \mathcal{D}_\mathcal{X}}\left\{ \mathrm{Tr}\left[H\rho(\boldsymbol{x})\right]\right\} \right| \leqslant \epsilon \right) \geqslant 1 - 2e^{-\frac{M\epsilon^2}{2}}.$$

Therefore, by the definition of average encoded state and linearity of trace operator:

$$\left| \mathrm{Tr}[H(\overline{\rho}_M)] - \mathrm{Tr}[H\mathbb{E}[\rho]] \right| \leqslant \epsilon$$

holds with probability at least $1 - 2e^{M\epsilon^2/2}$. $\qquad\square$

## G. More Experiments

In experiments, we employ the data re-uploading circuit as shown in the Fig. Figure G.1, where the parameterized gates consist of single-qubit rotation gates defined in Eq. (1) and two-qubit CNOT gates arranged in a ring connectivity pattern.

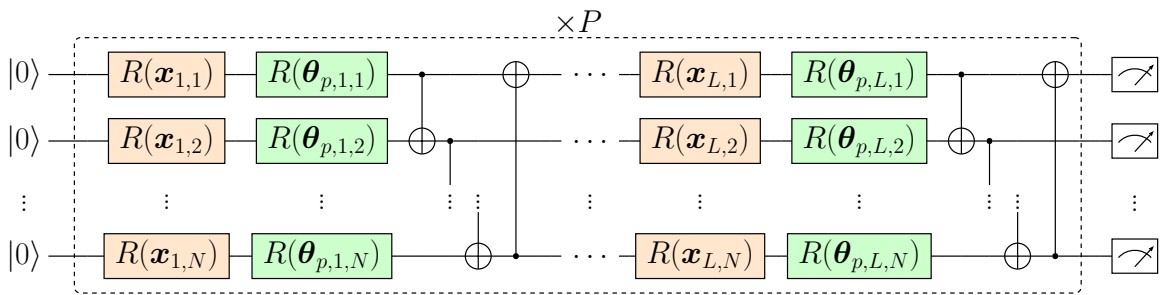

*Figure G.1.* Data re-uploading circuit used in experiments. The single qubit parameterized quantum gates are same with original data re-uploading circuit (Pérez-Salinas et al., 2020), and the entangling gates are CNOT gate with ring connectivity. Each parameter vector $\boldsymbol{\theta}_{p,l,n}$ is indexed by the repetition number $p$, encoding layer $l$, and qubit number $n$.

## G.1. Regression Experiments

**Data:** For regression tasks, we consider the multivariate function defined in $[-1, 1]^D$:

$$f(\boldsymbol{x}) = \frac{1}{2}\left(1 + \tanh\left(\sum_{i=1}^{D} x_i\right)\right), \tag{G.1}$$

where $D$ is the dimension of the data $\boldsymbol{x}$ and $f(\boldsymbol{x}) \in [0, 1]$. For each dimension $x_d$, we draw samples uniformly from $[-1, 1]$.

**Experimental Setup:** In experiments, we use two-qubit data re-uploading circuits as shown in Fig. G.1. Following the same setup as in Subsec. 4.2, to eliminate the impact of parameter count, we conduct comparative experiments using quantum circuits with fixed total layers $L_{\max} = 10$. For encoding layers $L < L_{\max}$, we maintain the total circuit layers at $L_{\max}$ by only encoding data in the first $L$ layers and setting the data input to zero vectors for the remaining layers (Using $L$ encoding layers). The observable used in experiments is $H_L = Z_1 \otimes Z_2$, where $Z_i$ is the Pauli-Z operator on the $i$-th qubit.

In experiments, the training set contains 600 samples, and the test set contains 10000 samples. Following the experimental setting in Subsec. 4.2, we initialize the circuit parameters $\boldsymbol{\theta}$ from normal distribution $\mathcal{N}(0, 1)$ and optimize using Adam with learning rate 0.005 and mini-batch size 200. The loss function is Mean Squared Error (MSE). Each experiment is trained for 1000 epochs and repeated 10 times with different random seeds. The results are shown in Fig. G.2.

**Results:** The experimental results are similar to those in Subsec 4.2. As shown in Fig. G.2 (d), as the number of encoding layers $L$ increases, the output with respect to the observable $H_L$ progressively approaches the output of maximally mixed state, leading to a corresponding increase in test error as shown in Fig. G.2 (c). According to Theorem E.2, the prediction error will be greater than $|\mathbb{E}_{\boldsymbol{x}\sim\mathcal{D}_\mathcal{X}}[f(\boldsymbol{x})]| - \epsilon$. For the function $f(\boldsymbol{x})$ defined in Eq. (G.1), its expected value is $\mathbb{E}_{\boldsymbol{x}\sim\mathcal{D}_\mathcal{X}}[f(\boldsymbol{x})] = 0.5$. Therefore, the prediction error will be greater than $0.5 - \epsilon$.

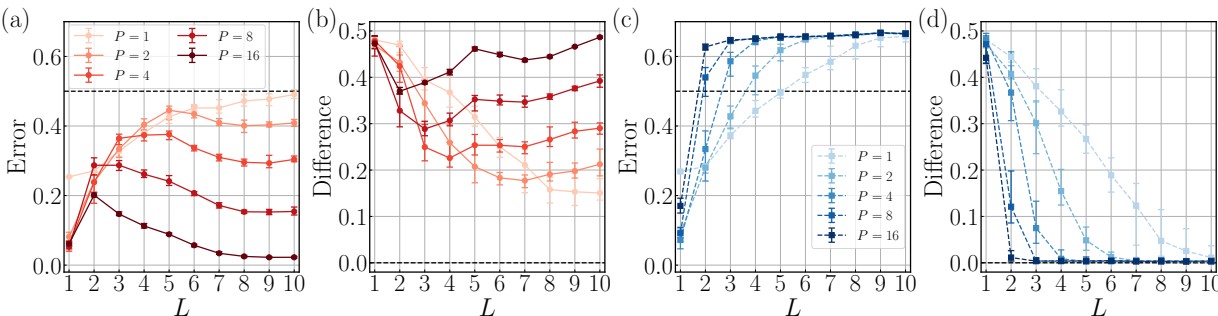

*Figure G.2.* (a) Training error; (c) Test error; (b) Difference between the output with respect to $H_L = Z_1 \otimes Z_2$ on training data and $\text{Tr}[H\rho_I]$. (d) Difference between the output with respect to $H_L$ on test data and $\text{Tr}[H\rho_I]$. Error bars represent the minimum and maximum values across 10 independent runs with different random seeds, with the central line showing the mean value.

## G.2. Impact of Dataset Size and Model Size

In traditional machine learning theory, prediction error is typically decomposed into two components: training error and generalization error (the gap between prediction error and training error). It is commonly believed that increasing model size can reduce training error but may increase generalization error, while increasing the size of the training dataset can reduce generalization error. The goal is to find a balance that achieves both low training error and low generalization error, thereby obtaining a small overall prediction error.

It is difficult to determine the prediction error solely by determining the training error and providing an upper bound on the generalization error. However, when the prediction error is determined, we can make the following inferences: under the same model complexity, increasing the training dataset size can reduce the generalization error, thereby increasing the training error; on the other hand, with the same training dataset size, increasing the model complexity can reduce the training error, but simultaneously increases the generalization error.

**Data:** We employed the same data as described in Subsec 4.3, with a feature dimension of $D = 24$.

**Experimental Setup:** To experimentally validate our inferences, we used a single-qubit data re-uploading circuit (Fig. G.1)

with encoding layer $L = 8$ (corresponding to the data dimension of 24). We investigated the effects of training set size and model complexity by systematically varying both the number of training samples and circuit repetitions. Other experimental settings followed those in Subsec. 4.2

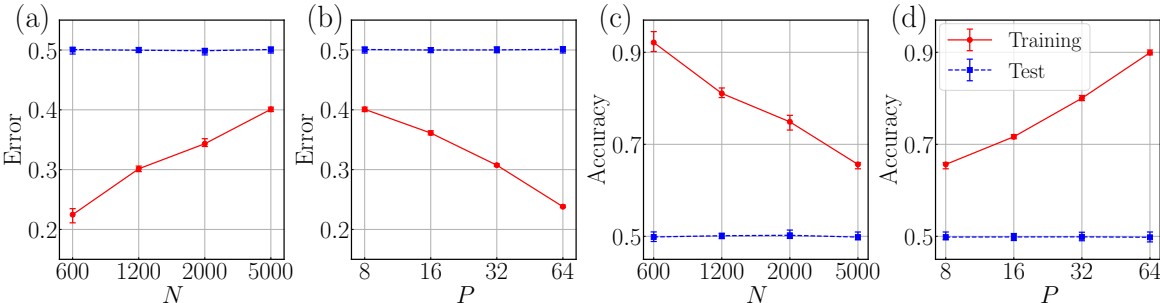

*Figure G.3.* (a) Training and test error under same model complexity but different training set size; (b) Training and test error under same training set size but different model complexity. (c) Training and test accuracy corresponding to (a); (d) Training and test accuracy corresponding to (b). Error bars represent the minimum and maximum values across 10 independent runs with different random seeds, with the central line showing the mean value.

**Results:** First, with fixed model complexity (repetition number $P = 8$), we varied the training set size as $[600, 1200, 2000, 5000]$. The training and test errors and accuracy are shown in Fig. G.3. We observe that as the training set size increases, the generalization error (gap between training error and test error) decreases while the test error remains constant, resulting in increasing training error.

Then, with fixed training set size of 5000, we increased model complexity by varying $P$ as $[8, 16, 32, 64]$. The training and test errors and accuracy are shown in Fig. G.3(b) and (d). We observe that the training error gradually decreases while the test error remains unchanged, and the generalization error increases.

### G.3. Accuracy Results

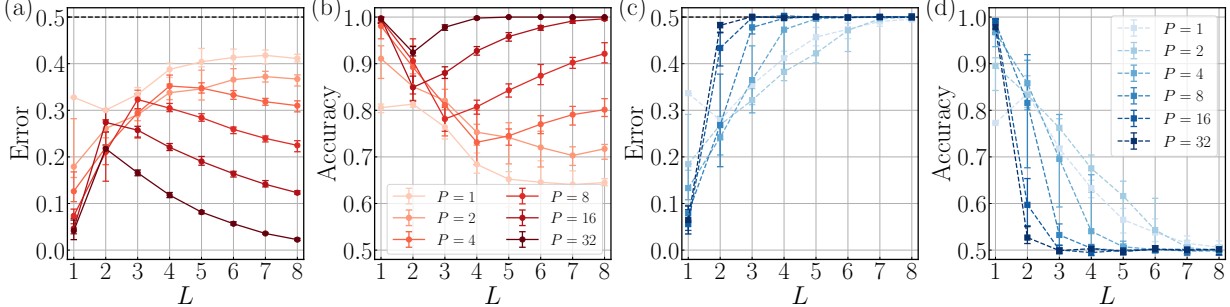

*Figure G.4.* (a) Training error from Fig. 5.(a); (b) Corresponding training accuracy for panel (a); (c) Test error from Fig. 6.(c); (d) Corresponding test accuracy for panel (c). Error bars represent the minimum and maximum values across 10 independent runs with different random seeds, with the central line showing the mean value.

In classification problems, accuracy is another commonly used evaluation metric alongside error. We present the training and test accuracy corresponding to the error results shown in the main text. Figure G.4 shows the accuracy corresponding to Figure 5, while Figure G.5 shows the accuracy for Figure 6. When the test error is around 0.5, the corresponding accuracy is also near 0.5, indicating random-guess level performance.

## H. Counter Example

In this appendix, we present a counter example that does not exhibit the limitations of data re-uploading with deep encoding layers as discussed in this paper. We demonstrate that this is due to the fact that the dataset essentially contains only few informative dimensions.

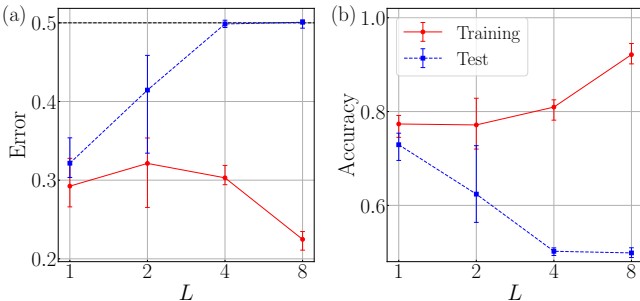

*Figure G.5.* (a) Training error from Fig. 6.(a); (b) Corresponding training accuracy for panel (a). Error bars represent the minimum and maximum values across 10 independent runs with different random seeds, with the central line showing the mean value.

**Data:** Consider a binary classification dataset where each sample $(\boldsymbol{x}^{(m)}, y^{(m)})$ consists of a $D$-dimensional feature vector $\boldsymbol{x}^{(m)} \in \mathbb{R}^D$ drawn from a Gaussian distribution $\mathcal{N}(\boldsymbol{\mu}, \boldsymbol{\Sigma})$. The mean vectors $\boldsymbol{\mu}$ are class-dependent: for class one, the mean of the $d$-th dimension is given by $\mu_d = \left[\frac{2\pi}{16}(d \bmod 8)\right] \bmod 2\pi$, while for class two, the mean of the $d$-th dimension is $\mu_d = \left[\frac{2\pi}{16}(8 + (d \bmod 8))\right] \bmod 2\pi$.

Both classes share the same covariance matrix $\boldsymbol{\Sigma} = \boldsymbol{\Lambda} + \boldsymbol{N}$, where $\boldsymbol{\Lambda}$ is a diagonal matrix with each diagonal element equal to 0.8, and $\boldsymbol{N}$ is a matrix with zero diagonal elements and all off-diagonal elements equal to $0.792(0.8 \times 0.99)$.

**Experimental Setup:** We employed the same experimental setup as in the Subsec. 4.2.

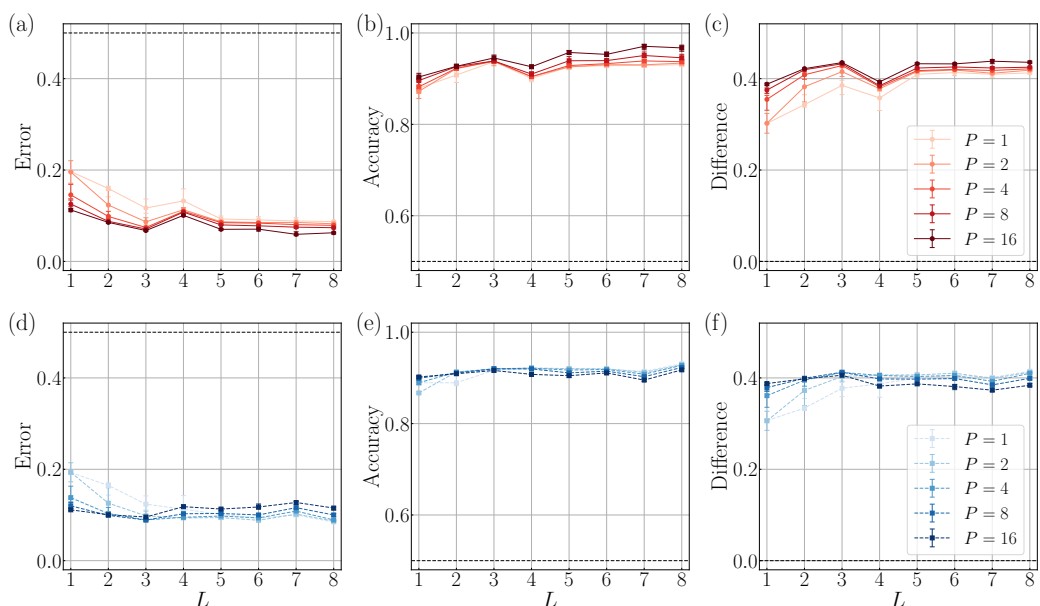

*Figure H.1.* (a) Training error; (b) Corresponding training accuracy; (c) Difference between the output with respect to $H_0 = |0\rangle \langle 0|$ on training data and $\mathrm{Tr}\,[H\rho_I]$; (d) Test error; (e) Corresponding test accuracy. (f) Difference between the output with respect to $H_0$ on test data and $\mathrm{Tr}\,[H\rho_I]$. Error bars represent the minimum and maximum values across 10 independent runs with different random seeds, with the central line showing the mean value.

**Results:** The experimental results are presented in Fig. H.1, showing that as the number of encoding layers $L$ increases, the test error does not approach 0.5. Both the test accuracy and training accuracy remain around 0.9, and the difference between the model's output on the test set and $\mathrm{Tr}\,[H\rho_I]$ stays around 0.4, without exhibiting the limitations of data re-uploading with deep encoding layers discussed in this paper.

In fact, by diagonalizing the covariance matrix $\boldsymbol{\Sigma} = \boldsymbol{Q}\boldsymbol{D}\boldsymbol{Q}^\top$ and applying a linear transformation $\boldsymbol{y} = \boldsymbol{Q}^T(\boldsymbol{X} - \boldsymbol{\mu})$ to the multivariate Gaussian random variable, we obtain a transformed variable $\boldsymbol{y}$ that follows a distribution $\mathcal{N}(\boldsymbol{0}, \boldsymbol{D})$. Here, $\boldsymbol{D}$ is

a diagonal matrix where only one element has significant magnitude while all other elements are negligible (0.008). To illustrate this, consider the case when $D = 24$: the variance of the dominant dimension is 19.16, whereas the variances of the remaining 23 dimensions are all 0.008. This indicates that the multivariate Gaussian data is essentially determined by a single dominant dimension (corresponding to the variance 19.16), while the other dimensions remain nearly constant.

The key insight is that while the data appears to be high-dimensional, it actually contains only one dominant dimension that carries meaningful information. This explains why the test error does not approach 0.5 even with increasing encoding layers, as the effective dimensionality of the data is much lower than its apparent dimensionality.

