# OpenReview forum: "Predictive Performance of Deep Quantum Data Re-uploading Models"
_ICML.cc/2025/Conference — ICML 2025 poster_

### Official Review · Reviewer_aAWZ · 2025-03-01

**Overall Recommendation:** 4

**Summary:**

This paper provides further theoretical insights into the reuploading approach of parameterized quantum circuits. They authors claim the divergence (and predictive error) are worse with increasing numbers of layers. The has implications for near term devices, since the number of qubits is much less than the dimensionality of the data often. Additional empirical results are shown to further highlight these claims.

**Claims And Evidence:**

In general, the claims are supported by evidence. The core claims of this paper are theoretical, and the experimental aspects demonstrate the expect effects (but are not necessary to prove the theory is true).

**Essential References Not Discussed:**

In general, the references are sufficient, one that would be worth adding is https://arxiv.org/abs/2501.16228 in which the effect of e.g. number of reuploading layers is considered in terms of generalization capability of the model.

**Experimental Designs Or Analyses:**

Yes. The experiments are generally valid. However, figure 4 it looks like the pre-training is missing/overlapping and is hard to identify. Also for Figure 5, it would be worth having the error bars (like in later figures) to give an estimate of if the variance is important.

**Methods And Evaluation Criteria:**

The empirical evaluation benchmarks are fine. The artificial datasets highlight the points made by the theory and the later CIFAR/MNIST highlight more realistic datasets.

**Other Comments Or Suggestions:**

A few minor typographic things could be improved (e.g. Fig G1 gets reference but the link when clicked on goes to Figure 1 not Figure 1 of appendix G). Additionally in Figure 7, I think it would benefit from more color scaled lines showing different repetitions as was done in figure 5. Figure 6 would benefit from more datapoints (adding more points to the line).

**Other Strengths And Weaknesses:**

The paper is overall clear and to the point. The theory is clearly stated, and the results highlight expected features of the theory. One point that could be clearer is the impact of the assumptions about the data. Specifically, in 3.1 data is drawn from a Gaussian for each point. Although this allows the Pauli decomposition, it would be beneficial to further discuss the implications/strength of this assumption.

**Questions For Authors:**

1. In 4.3, what is the need to keep entanglement effects out? Does this meaningfully change the trend/statistics?

**Relation To Broader Scientific Literature:**

The results connect to the broader scientific body of literature in several ways. First, it builds upon the generally large body of literature on trainability of quantum circuits. This paper highlights the difficulties in training deep reuploaded circuits, as many other papers have shown the difficulty of training circuits like HEAs, or shown the difficulties in training circuits with circuit cost functions (e.g. local vs global). Second, it builds specifically upon the body of work analyzing the reuploading circuit. The reuploading approach is quite common in QML works because it is useful to encode data larger than the number of qubits (which is common in hardware, and all but ubiquitous in simulation), and has built quite a substantial influence in the QML community. Previous works have indicated difficulties in training/expressivity of reuploading circuits (as pointed out in the paper) and this expands in that direction.

**Theoretical Claims:**

I did not check the correctness of proofs that were not in the main body.

---

> ### Author Rebuttal · Authors · 2025-03-30
>
> We greatly appreciate the reviewer's thorough and constructive feedback, which has helped us improve the clarity and quality of our manuscript. We address each point below.
>
> ## figure 4 it looks like the pre-training is missing/overlapping and is hard to identify.
>
> We thank you for pointing out this issue that was not clarified in the figure caption. Figure 4 is organized into left panels (a,c), showing pre-training (before training) divergence results marked by circles, and right panels (b,d), showing post-training (after training) divergence results marked by squares. We will update the caption of Figure 4 in the next version.
>
>
> ## for Figure 5, it would be worth having the error bars to give an estimate of if the variance is important.
>
> We appreciate this valuable suggestion. We have added error bars to the figure. The maximum, minimum, and mean prediction errors in 10 independent runs consistently demonstrate that as the number of encoding layers increases, the predictive performance approaches random guessing. Additionally, as the repetition number $P$ increases, the variance gradually decreases. We will update the figure with error bars in the next version.
>
>
> ## one that would be worth adding is https://arxiv.org/abs/2501.16228.
>
> We thank you for bringing this relevant reference to our attention. We will cite and add it to the related work section in the next version.
>
> ## One point that could be clearer is the impact of the assumptions about the data. Specifically, in 3.1 data is drawn from a Gaussian for each point. Although this allows the Pauli decomposition, it would be beneficial to further discuss the implications/strength of this assumption.
>
> We adopt the assumption of Gaussian distribution because of its prevalence and convenience for proof. In fact, our conclusions are not limited to Gaussian distributions. The role of the Gaussian distribution in our proof is mainly reflected in Lemma C.4 of Appendix C. As long as the distribution of data $x$ satisfies $\mathbb{E}[\cos(x)] = \gamma \cos (\mu)$, where $|\gamma| < 1$, our main point still holds. For example, when $x$ follows a uniform distribution over $[\mu-a,\mu+a]$, we have $\mathbb{E}[\cos(x)] = \sin(a)/ a \cdot \cos(\mu)$, where $|\gamma| = |\sin(a) / a| < 1$ when $a \neq 0$.
>
>
> We will elaborate on this point in more detail in the Discussion section of the next version.
>
>
> ## Fig G1 gets reference but the link when clicked on goes to Figure 1 not Figure 1 of appendix G
>
> Thank you for your careful attention to detail. We have fixed this issue.
>
>
> ## in Figure 7, I think it would benefit from more color scaled lines showing different repetitions
>
> We appreciate this constructive suggestion. Building upon our original experiments (data re-uploading model with repetition $P=2$), we have added more experiments with $P=4$ and $P=8$, and updated Figure 7 using color scaled lines. The results continue to show that MNIST has the highest test accuracy (fewer encoding layers, smaller data variance), followed by grayscale CIFAR10 (fewer encoding layers), while RGB CIFAR10 has the worst test accuracy, consistently around 0.5 (deeper encoding layers). Furthermore, as the number of repetitions increases, the training accuracy improves while the test accuracy decreases, indicating larger generalization error. This aligns with the findings reported in https://arxiv.org/abs/2501.16228. We will include this updated Figure 7 with color scaled lines in the next version of our manuscript.
>
>
> ## Figure 6 would benefit from more datapoints (adding more points to the line).
>
> Thank you for this thoughtful suggestion. For Figure 6, our experiments used fixed data dimensionality ($D = 24$) requiring 8 encoding gates (since each encoding gate can encode three classical data elements). We have already explored all possible configurations under the same parameter settings: 8 qubits with 1 encoding layer, 4 qubits with 2 encoding layers, 2 qubits with 4 encoding layers, and 1 qubit with 8 encoding layers. While adding more data points isn't feasible given these experimental constraints, we sincerely appreciate your careful review and valuable suggestion for improving our work.
>
>
> ## In 4.3, what is the need to keep entanglement effects out? Does this meaningfully change the trend/statistics?
>
> We thank you for this thoughtful question regarding our experimental methodology. We chose to exclude entanglement effects from our experimental design because prior studies [1,2] has shown that entanglement can significantly degrade both training and prediction performance. By eliminating this confounding factor, we were able to isolate and study specifically how the number of encoding layers impacts model behavior.
>
>
> [1] Ortiz Marrero, C., Kieferová, M. and Wiebe, N., 2021. Entanglement-induced barren plateaus. PRX quantum, 2(4), p.040316.
>
> [2] Leone, L., Oliviero, S.F., Cincio, L. and Cerezo, M., 2024. On the practical usefulness of the hardware efficient ansatz. Quantum, 8, p.1395.

---

> > ### Comment · Reviewer_aAWZ · 2025-04-02
> >
> > I thank the authors for their amenable response. I believe with the many changes they are making to figure and explanations as outlined in their response (in addition to the changes from other reviewers responses), the paper will be improved. As such, I have updated my score.

---

> > > ### Author Response · Authors · 2025-04-09
> > >
> > > Thank you for acknowledging our work and raising the score. We truly appreciate your time and effort in reviewing our paper.

---

### Official Review · Reviewer_QK5P · 2025-03-14

**Overall Recommendation:** 3

**Summary:**

The paper investigates the effectiveness of quantum machine learning models that use data re-uploading circuits.
These models have gained attention for their expressivity and trainability, but their ability to make accurate predictions on unseen data remains under-explored.
The study highlights a limitation in deep quantum data re-uploading models and provides guidance for designing better quantum machine learning architectures.
It suggests that increasing the depth does not necessarily improve model performance, especially when dealing with high-dimensional classical data.
Instead, wider quantum circuits may be the key to more effective quantum learning models.

**Claims And Evidence:**

- The paper provides theoretical proof showing that as encoding layers increase, the encoded quantum states converge toward a maximally mixed state, leading to predictions that approach random guessing.

- Theoretical analysis shows that increasing the number of repetitions cannot reduce the loss of distinguishability caused by deep encoding layers. Experimental results confirm that increasing repetitions does not improve the performance.

- The study presents comparative experiments using circuits with different numbers of encoding layers but the same total parameter count. The experimental results show that circuits with fewer encoding layers generalize better.

**Essential References Not Discussed:**

There are no additional related works that are essential to understanding the key contributions of the paper.

**Experimental Designs Or Analyses:**

As mentioned in "Methods And Evaluation Criteria", the paper uses synthetic datasets to evaluate the theoretical claims.
The paper also uses real datasets to evaluate the limitations in practical scenarios.

**Methods And Evaluation Criteria:**

- The paper provides mathematical analysis between the quantum states and a maximally mixed state, which is intuitive to show the performance of the models when the depth of the encoding is increased.

- The paper uses both synthetic and real-world datasets to show the effect of encoding depth and theoretical limitations in practical scenarios.

**Other Comments Or Suggestions:**

No other comments or suggestions.

**Other Strengths And Weaknesses:**

What I am concerned about is the originality and novelty of the theorems 3.1 and 3.2.
This is because these theorems are mentioned and proved in (Li et al. 2022) (see Eqn (5) in Theorem 2 and Eqn (7) in Corollary 2.1).
Although the paper notes that the contributions are expanded from (Li et al. 2022), the novelty should be emphasized in the main paper.

**Questions For Authors:**

I think in the main paper, the authors should highlight the differences between the proof from (Li et al. 2022) and the authors' paper.
This could make the reader less confused since the paper claims that the limitation can be proved in a broader case.

**Relation To Broader Scientific Literature:**

The contributions of the paper can help to understand the expressivity in data encoding designs.

**Theoretical Claims:**

The proofs of the theoretical claims are checked. The proofs are based on the assumption that the input features are independent Gaussian-distributed, which is acceptable.

---

> ### Author Rebuttal · Authors · 2025-03-30
>
> We thank you for the thorough and insightful comments, which helped us improve the clarity and depth of our manuscript. Below we elaborate on the differences between our work and (Li et al. 2022) in terms of overall framework and proof techniques.
>
> ## Framework for analyzing prediction error
>
> Regarding the overall framework, we introduce a novel method to analyze the prediction error, bypassing the traditional decomposition into training and generalization errors. We directly analyze the expected output of the model over the data distribution. Our results demonstrate that when using data re-uploading models with deep encoding layers, the model's performance on unseen new data approaches random guessing, regardless of training quality, loss function choice, optimization method (gradient-based or gradient-free), iteration count, number of parameters, or training sample size.  In contrast, the results in (Li et al. 2022) are limited to showing that the gradient is small for quantum machine learning models when optimizing with cross entropy loss function.
>
> Indeed, directly analyzing prediction error, rather than decomposing it into training and generalization errors, is important. As shown in Appendix G.2, traditional machine learning theory suggests that increasing model complexity reduces training error while increasing generalization error, and increasing training sample size increases training error while reducing generalization error. However, since both components change dynamically, it is challenging to theoretically determine their sum (the prediction error). Our analysis method confirms that regardless of how training and generalization errors vary, the prediction error consistently approaches random guessing.
>
>
> ## Proof techniques
>
> Regarding proof techniques, when analyzing the expected output of the model over the data distribution, our analysis focuses on two key aspects: the impact of the number of encoding layers and the number of repetitions. For analyzing the number of encoding layers, we employ techniques similar to (Li et al. 2022). However, Li's results (Theorem 2) only allow specific non-parameterized entangling gates (CNOT or CZ) between encoding layers. Our results (Theorem 3.1 in the main paper) allow arbitrary learnable parameterized quantum gates between encoding layers (Proof in Appendix C).
>
> Furthermore, Li's techniques cannot analyze scenarios involving repeated data uploading. Corollary 2.1 in Li's paper, directly derived from their Theorem 2, only examines the relationship between the model's expected output and maximally mixed states **without** repeated data uploading. In contrast, our result (Theorem 3.2 in the main paper) cannot be directly derived from Theorem 3.1; instead, it establishes the relationship between the model's expected output and maximally mixed states **with** repeated data uploading through a non-trivial extension. We address this limitation by constructing approximating circuits to analyze such cases (proof provided in Appendix D), a contribution absent in Li's paper. Importantly, repeated data uploading is crucial because it significantly enhances the trainability of these models[1].
>
>
> ## Overall
>
> We thank the reviewer for prompting us to clarify these important distinctions. In fact, the final result in (Li et al. 2022) (Proposition 4) highlights the trainability issue in quantum machine learning models lacking repeated data uploading and learnable parameters between encoding layers. The data re-uploading paradigm addresses these limitations by incorporating two key elements: learnable parameter gates between encoding layers and repeated data uploading[1,2,3]. The improved trainability of data re-uploading models is a key factor driving their widespread adoption in the field.
>
>
> Our theory presents a novel perspective: even though data re-uploading models with deep encoding layers may exhibit good trainability, their predictive performance approaches random guessing. This phenomenon has not been previously discovered, as earlier works [1,2,3] and (Li et al. 2022) primarily focused on model trainability.
>
> We appreciate the opportunity to improve our manuscript and will revise the Contributions section in the main paper to clearly highlight our contributions and the key differences between our proof techniques and those in (Li et al. 2022).
>
>
>
>
>
>
>
> [1] Pérez-Salinas, A., Rad, M.Y., Barthe, A. and Dunjko, V., 2024. Universal approximation of continuous functions with minimal quantum circuits. arXiv preprint arXiv:2411.19152.
>
> [2] Pérez-Salinas A, Cervera-Lierta A, Gil-Fuster E, Latorre JI. Data re-uploading for a universal quantum classifier. Quantum. 2020 Feb 6;4:226.
>
> [3] Yu, Z., Chen, Q., Jiao, Y., Li, Y., Lu, X., Wang, X. and Yang, J.Z., 2023. Provable advantage of parameterized quantum circuit in function approximation (No. arXiv: 2310.07528).

---

> > ### Comment · Reviewer_QK5P · 2025-04-07
> >
> > I thank the authors for the clarification. I have updated the score.

---

> > > ### Author Response · Authors · 2025-04-09
> > >
> > > Thank you for acknowledging our work and raising the score. We truly appreciate your time and effort in reviewing our paper.

---

### Official Review · Reviewer_1fvX · 2025-03-16

**Overall Recommendation:** 3

**Summary:**

The authors examine the predictive performance of data re-uploading models, a class of variational quantum circuits which has attracted significant attention in recent years. They prove that under certain theoretical assumptions about the data generating distribution, these models' have the property that as one increases the numebr of data encoding layers, the average output becomes indistinguishable from a maximally mixed state.  This property can be used to give lower bounds on the predictive accuracy of such models. The authors also validate their findings with experiments on both synthetic and real world datasets. They argue that their results imply that data reuploading models should be designed with large width rather than large depth.
##
Update after rebuttal: I would like to thank the authors for their responses. I have decided to maintain my score.

**Claims And Evidence:**

The authors' claims regarding the limitations of data re-uploading models are thought provoking, and their presentaion is clear, but I am not fully convinced that they imply that deep data re-uploading models cannot be made to work in practice, perhaps with some tweaks analogous to those used in the classical ML literature for deep models (residual connections, batch normalization etc.).

**Essential References Not Discussed:**

I am not aware of any such references.

**Experimental Designs Or Analyses:**

I did not check the soundness of the experimental results.

**Methods And Evaluation Criteria:**

The methods use seem appropriate for the problem at hand.

**Other Comments Or Suggestions:**

I have no other comments.

**Other Strengths And Weaknesses:**

I believe the paper represents a solid contribution to the theory of data reuploading models- its results are certainly thought provoking, and should spark discussion about their implications for practice.

**Questions For Authors:**

1.The results suggest that any data re-uploading model  (in the regime studied by the authors, and with gaussian inputs etc) will be roughly speaking "mean zero". Could one not add a clasical bias term which will be trained alongside the data reuploading model to mitigate this?
2. The authors study a regime where the dimensionality of the data grows linearly with L. Presumably the results would not hold if the dimensionality of the data were fixed?
3. In classical ML, one can also show that taking the number of layers of a NN to infinity, at least if one does this naively, results in pathological behavior- how does the phenomena decribed in this work compare to that?

**Relation To Broader Scientific Literature:**

The authros provide a solid contribution to the theoretical undertanding of data re-uploading models, a class of variational quantum circuits which has received widespread attention.

**Theoretical Claims:**

I did not check the correctness of the proofs.

---

> ### Author Rebuttal · Authors · 2025-03-30
>
> We thank you for the insightful questions and valuable feedback. Our responses to your questions are as below.
>
> ## 1. The results suggest that any data re-uploading model (in the regime studied by the authors, and with gaussian inputs etc) will be roughly speaking "mean zero". Could one not add a classical bias term which will be trained alongside the data reuploading model to mitigate this?
>
> We thank you for this constructive suggestion. We thought of this possibility, and found that adding a classical bias term would improve the model's overall predictive performance to some extent, but the data re-uploading model has no contribution to the improvement. In other words, when the data re-uploading model is in a "mean zero" state, this model becomes informationless, and the model's prediction is solely dependent on the information provided by the classical bias term for classification and regression.
>
> ## 2. The authors study a regime where the dimensionality of the data grows linearly with L. Presumably the results would not hold if the dimensionality of the data were fixed?
>
> This is an important question to be clarified. Our answer is yes. Actually, in Section 4.3 (Classification Experiments in Same Dataset) of the main paper, we conduct experiments using the same dataset with fixed data dimensionality ($D = 24$). Encoding data of this dimensionality into quantum circuits requires 8 encoding gates (since each encoding gate can encode three classical data elements). We adjust the number of encoding layers by varying the width of quantum circuits (number of qubits).
>
> We investigate four different configurations: 8 qubits with 1 encoding layer, 4 qubits with 2 encoding layers, 2 qubits with 4 encoding layers, and 1 qubit with 8 encoding layers. Figure 6 shows that as the number of encoding layers increases, the model's predictive performance gradually degenerates to the level of random guessing (with prediction error approaching 0.5). Additionally, the model's average output on the test set converges towards a maximally mixed state, indicating that the model becomes informationless.
>
> ## 3. In classical ML, one can also show that taking the number of layers of a NN to infinity, at least if one does this naively, results in pathological behavior-how does the phenomena described in this work compare to that?
>
> We thank you for raising this interesting comparison with classical neural networks. It should be pointed out that the number of layers in classical neural networks and data re-uploading models generally refer to different concepts. In classical neural networks, the number of layers typically refers to parametric layers, while in data re-uploading models considered in this paper, the number of layers refers to encoding layers that are used to encode data into the quantum circuit. Classical neural networks input data all at once and continuously learn from it through parametric layers, whereas data re-uploading models partition the data and encode it through encoding layers, which allows encoding high-dimensional data using a limited number of qubits.
>
> A comparable quantum machine learning paradigm to classical machine learning is the encoding-variational paradigm, where data is first encoded into the quantum circuit all at once and then learned through parametric layers. Similar to classical machine learning, pathological behavior emerges as the number of parametric layers approaches infinity. In the quantum machine learning community, this pathological behavior is known as the Barren Plateau phenomenon, where the gradient of the loss function with respect to the model parameters vanishes as the number of parametric layers becomes too large, causing severe trainability issues, as discussed in reference [1]. The data re-uploading paradigm addresses this by placing parametric layers between encoding layers, potentially alleviating the trainability problem. However, the predictive performance of data re-uploading model remains unexplored, which is the central question we aim to answer in this paper.
>
>
>
> [1] Cerezo, M., Sone, A., Volkoff, T., Cincio, L. and Coles, P.J., 2021. Cost function dependent barren plateaus in shallow parametrized quantum circuits. Nature communications, 12(1), p.1791.

---

### Decision · Program_Chairs · 2025-05-01

**Decision:**

Accept (poster)

**Comment:**

This submission investigates the effectiveness of quantum machine learning models with data re-uploading, offering insights into which quantum circuit architectures may be more favorable for quantum learning tasks. All reviewers agree that this is an important and solid contribution. The authors are encouraged to consider the reviewers' comments on presentation in their revision.